# Learning to Memorize with Attributive and Associative Memory for Online Test-Time Adaptation of Vision-Language Models

Yuchao Zhang [1 2]  Hao Wang [3]  Fan Zhang [4]  Qirui Mi [5]  Mengyue Yang [6]  Yisen Wang [3]  Jun Wang [7]
Haoxuan Li[✉ 1]  Zhouchen Lin[✉ 3]

## Abstract

Memory-based test-time adaptation (TTA) assigns streaming test samples into class-specific memory slots based on pseudo-labels predicted by models like CLIP, and retrieves them to facilitate subsequent predictions under distribution shift. However, this process introduces two challenges: ❶ **Each sample is hard-assigned to a single class based on CLIP's prediction**, where inaccurate CLIP prediction leads to memory contamination that biases subsequent prediction. ❷ **Samples are evicted under biased selection due to fixed memory capacity**, which risks discarding informative samples and undermining the efficacy of the memory. To address these challenges, we propose $A^2$**Memory** (**A**ttributive-**A**ssociative **Memory** for Test-time Adaptation). For challenge ❶, we propose *Attribute-centric Memory Construction* that builds prior textual representations from class-shared representative and diverse visual attributes, and applies soft assignment to generate surrogate visual representations. For challenge ❷, we design *Class-wise Associative Memory* that dynamically compresses streaming samples into fixed-capacity memory through gradient-based optimization and data-dependent retention, then retrieves sample-adaptive class prototypes for reliable inference. Extensive experiments demonstrate consistent improvements over state-of-the-art methods across 15 benchmarks.

## 1. Introduction

Test-time adaptation (TTA) is a critical technique for enhancing the generalization of vision-language models (VLMs) to out-of-distribution scenarios without requiring labeled target data (Shu et al., 2022; Zhang et al., 2024d). Specifically, VLMs such as CLIP (Radford et al., 2021) learn well-aligned visual–text representations through large-scale pre-training, yet they yield suboptimal performance at deployment due to distribution shifts, such as changes in background context (Zhou et al., 2022b;a). TTA addresses this by adapting model predictions using only unlabeled test samples as they arrive, bridging the gap between pre-training and target distributions. Early TTA methods for VLMs rely on prompt tuning (Shu et al., 2022; Yoon et al., 2024; Xiao et al., 2025), adaptively adapting domain-specific prompts at inference via entropy minimization. However, their cost is high due to backpropagation through the full encoder, which limits efficiency in online streaming under strict latency constraints.

To address this efficiency bottleneck, memory-based methods have gained prominence in online adaptation due to their lightweight computation and competitive performance (Karmanov et al., 2024; Zhang et al., 2024b;a; Chen et al., 2025b; Guan et al., 2026; Dai & Yang, 2025). These approaches maintain a memory that stores streaming test samples, assign each sample to a class based on its pseudo-label predicted by CLIP, and refine predictions via similarity-weighted retrieval. This line of work is initiated by TDA (Karmanov et al., 2024), which proposes positive-negative dual memories with entropy-based filtering. Subsequent works improve it through diverse methods such as token-level memory condensation (Wang et al., 2025), hybrid memory architectures (Zhang et al., 2024d), and multi-level feature aggregation (Chen et al., 2025b). Building on these advances, memory-based methods have become a mainstream paradigm for VLM test-time adaptation.

Despite these advances, two fundamental limitations persist in sample-level memory-based methods. ❶ **Each sample is hard-assigned to a single class based on CLIP's prediction**, where incorrectly assigned samples contaminate the memory, particularly at early stages when most incom-

---

[1]Institute for Artificial Intelligence, Peking University [2]Beijing University of Chemical Technology [3]State Key Lab of General AI, School of Intelligence Science and Technology, Peking University [4]The Chinese University of Hong Kong [5]Key Laboratory of Interdisciplinary Research of Computation and Economics, Shanghai University of Finance and Economics [6]University of Bristol [7]University College London. Correspondence to: Haoxuan Li <hxli@stu.pku.edu.cn>, Zhouchen Lin <zlin@pku.edu.cn>.

*Proceedings of the 43rd International Conference on Machine Learning*, Seoul, South Korea. PMLR 306, 2026. Copyright 2026 by the author(s).

ing samples are retained due to sufficient capacity without filtering. This biases subsequent retrieval and impairs the reliability of the memory during later prediction. ❷ **Samples are evicted under biased selection due to fixed memory capacity**. When capacity is exceeded, eviction strategies discard potentially informative samples, introducing selection bias in the retained memory. This sample-level storage paradigm limits memory expressiveness by losing discriminative information from evicted ones.

To address these challenges, we propose $\mathbf{A}^2\mathbf{Memory}$ (**A**ttributive-**A**ssociative **Memory** for Test-time Adaptation), a framework that reformulates memory-based TTA as attribute-centric associative memory optimization (Yang et al., 2023; Behrouz et al., 2025b;a), where the memory is parameterized as class-wise key–value mappings guided by shared visual attributes and learned through gradient-based updates (Yang et al., 2024a;b), rather than sample-level storage paradigm. For ❶, we introduce *Attribute-centric Representation Construction*, which selects representative and diverse visual attributes shared across classes to build attribute-level prior textual representations, and applies attribute-centric soft assignment to generate surrogate visual representations, mitigating memory contamination caused by hard class assignments. For ❷, we design *Class-wise Associative Memory*, which treats surrogate visual representations as keys and prior textual representations as values, updating it via gradient-based optimization and data-dependent retention to retrieve sample-adaptive class prototypes for reliable inference. This formulation enables dynamic compression of streaming samples into fixed-capacity memory while preserving discriminative information.

**Contributions.** The contributions of this work can be summarized as follows. ❶ **We identify two fundamental limitations in existing memory-based test-time adaptation methods:** hard pseudo-label assignment leads to memory contamination that biases subsequent retrieval, and heuristic sample eviction under fixed capacity introduces selection bias and limits memory expressiveness. ❷ **We propose $\mathbf{A}^2\mathbf{Memory}$ for the adaptation of the test-time**, which integrates Attribute-Centric Representation Construction for uncertainty-aware soft assignment with Class-wise Associative Memory for retrieving sample-adaptive prototypes from key-value mappings. ❸ **We evaluate the effectiveness of $\mathbf{A}^2\mathbf{Memory}$ across 15 benchmarks**, demonstrating consistent improvements over state-of-the-art methods while maintaining computational efficiency.

## 2. Preliminary

### 2.1. CLIP for Zero-Shot Classification

CLIP (Radford et al., 2021) aligns visual and textual representations via contrastive pre-training. It comprises an image encoder $f_\theta$ and a text encoder $g_\psi$. For a $C$-class task, class names are embedded into prompt templates (e.g., "*a photo of a [class]*") to generate prototypes $\boldsymbol{w}_c = g_\psi(t_c)$. Given a test image $\boldsymbol{x}^t$ with feature $\boldsymbol{f}^t = f_\theta(\boldsymbol{x}^t)$, the zero-shot prediction is computed via cosine similarity $\langle \cdot, \cdot \rangle$:

$$p(y = c \mid \boldsymbol{x}^t) = \frac{\exp(\langle \boldsymbol{f}^t, \boldsymbol{w}_c \rangle / \tau)}{\sum_{k=1}^{C} \exp(\langle \boldsymbol{f}^t, \boldsymbol{w}_k \rangle / \tau)}, \quad (1)$$

where $\tau$ is the temperature. The prediction is $\hat{y} = \arg\max_c p(y = c \mid \boldsymbol{x}^t)$.

### 2.2. Memory-based Test-Time Adaptation

Standard memory-based TTA methods (Karmanov et al., 2024; Zhang et al., 2022) adapt VLMs by maintaining a *non-parametric* memory $\mathcal{M} = \{\mathcal{M}_c\}_{c=1}^C$, where streaming test samples are assigned to class-specific slots $\mathcal{M}_c$ based on the pseudo-labels predicted by Eq. (1). Each $\mathcal{M}_c = \{\boldsymbol{f}_{c,i}\}_{i=1}^M$ stores historical visual features with fixed size $M$, updated by replacing samples with those exhibiting lower prediction entropy of Eq. 1 to retain confident ones. At inference, for a test sample $\boldsymbol{f}^t$, the model utilizes $\mathcal{M}_c$ to compute a retrieval-based logit $p_c^{mem}$ via similarity-weighted aggregation:

$$p_c^{mem} = \sum_{\boldsymbol{f}_{c,i} \in \mathcal{M}_c} \exp\big(-\beta(\boldsymbol{f}^t)^\top \boldsymbol{f}_{c,i}\big), \quad (2)$$

where $\beta$ is a hyperparameter. The final prediction fuses the logits in Eq. (1) with $p_c^{mem}$. However, this explicit storage paradigm is constrained by fixed capacity and heuristic replacement strategies, which limits feature expressiveness.

### 2.3. Associative Memory

Associative memory (Behrouz et al., 2025b;a) compresses sequential interactions into a fixed-size *parametric* matrix $\mathbf{M}^t \in \mathbb{R}^{D \times D}$, offering a continuous alternative to discrete sample storage. Given feature $\boldsymbol{z}^t$ at step $t$, it projects the feature into queries $\boldsymbol{q}^t$, keys $\boldsymbol{k}^t$, and values $\boldsymbol{v}^t$ to update the memory. The retrieval produces an output $\boldsymbol{o}^t$ by querying memory:

$$\boldsymbol{o}^t = \mathbf{M}^t \cdot \phi(\boldsymbol{q}^t). \quad (3)$$

Crucially, $\boldsymbol{o}^t$ represents the associative recall, which is the representation most relevant to the current query $\boldsymbol{q}^t$ reconstructed from the compressed history. The memory $\mathbf{M}^t$ is updated recursively to minimize the reconstruction error of the current key-value pair while retaining past knowledge:

$$\mathbf{M}^t = \arg\min_{\mathbf{M}} \|\mathbf{M}\phi(\boldsymbol{k}^t) - \boldsymbol{v}^t\|_2^2 + \|\mathbf{M} - \mathbf{M}^{t-1}\|_F^2. \quad (4)$$

It is solved via gradient descent (Yang et al., 2024a), allowing the memory to dynamically adapt to new patterns (plasticity) while preserving learned associations (stability).

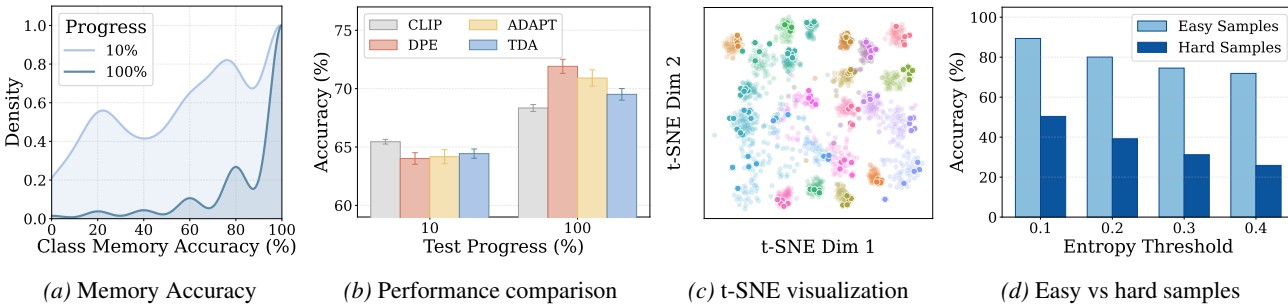

*(a)* Memory Accuracy      *(b)* Performance comparison      *(c)* t-SNE visualization      *(d)* Easy vs hard samples

*Figure 1.* **Analysis of memory-based TTA on ImageNet with ViT-B/16.** (a) Memory accuracy distribution in two stages. (b) Accuracy of memory-based methods in two stages. (c) Sample distribution under entropy-based selection. Dark points indicate retained samples; colors denote classes. (d) Accuracy of easy (high-confidence) and hard (low-confidence) samples across entropy thresholds.

## 3. Methodology

### 3.1. Motivation

Memory has become a prevalent approach to the test-time adaptation of modern vision-language models. Memory-based approaches offer computational efficiency by storing test features with pseudo-labels and refining predictions through similarity-weighted retrieval (Zhang et al., 2022; Karmanov et al., 2024). The memory serves as an auxiliary knowledge base capturing target domain statistics, making the quality of the memory crucial for successful adaptation.

Despite their practical appeal, existing methods suffer from two fundamental limitations inherent to memory-based test-time adaptation. ❶ **Each sample is hard-assigned to a single class based on CLIP's prediction.** Test samples are routed to a single class memory via CLIP pseudo-labels. When CLIP accuracy degrades, misassignments contaminate memory slots and bias subsequent retrieval and prediction. ❷ **Samples are evicted under biased selection due to fixed memory capacity.** Once memory capacity is exceeded, eviction strategies discard samples to admit higher-confidence ones, introducing selection bias and limiting memory expressiveness to a narrow subset.

We substantiate these limitations through empirical analysis. As seen in the discrepancy between 10% and 100% progress in Fig. 1a, memory accuracy remains low in the early stages due to the unfiltered retention of incorrect pseudo-labels under sufficient capacity. Here, *class memory acc* measures the fraction of correctly pseudo-labeled samples in class memory $\mathcal{M}_c$, i.e., $\frac{1}{|\mathcal{M}_c|} \sum_{x \in \mathcal{M}_c} \mathbf{1}[\hat{y}(x) = y(x)]$, and *density* denotes the relative frequency of class memories falling into each accuracy bin, normalized to $[0, 1]$. Consequently, Fig. 1b demonstrates that memory-based methods even underperform zero-shot CLIP during early adaptation, indicating that contaminated memory introduces biased rather than effective guidance for prediction. Regarding capacity constraints, Fig. 1c visualizes pronounced sample selection bias showing that samples retained by entropy-based evic-

tion concentrate in a narrow region of the feature space and fail to cover the full test distribution, which limits memory expressiveness. As a result, Fig. 1d shows that sample selection mainly improves performance on easy samples (where $\tilde{H}(\mathbf{P}^{clip}) < \tau$) while providing limited benefit for hard ones ($\tilde{H}(\mathbf{P}^{clip}) \geq \tau$, with $\tau$ a fixed visualization cutoff), indicating biased memory guidance is insufficient for challenging samples.

### 3.2. Attribute-centric Representation Construction

**Margin-gain Based Attribute Search.** In this section, we introduce *Prior Textual Representation* to provide attribute-level semantic guidance for memory-based test-time adaptation. We first generate a set of class-shared visual attributes using a Large Language Model (Achiam et al., 2023) and elicit attribute-level textual representations for each class, and then select a compact subset of representative and diverse attributes via marginal-gain-based selection to form prior textual representation.

We define a shared set of $N$ candidate visual attributes that are *class-shared* where the same attribute vocabulary applies across all categories. For each class $c$, we prompt an LLM to instantiate these shared universal attributes to generate $N$ candidate descriptions $\{t_{c,n}\}_{n=1}^{N}$ (details in Appendix C.1), where $t_{c,n}$ describes class $c$ from the perspective of shared-attribute $n$. Each description is encoded via CLIP's text encoder to derive candidate prior textual representation $\mathbf{T}_{c,n}^{can} = \text{norm}(g_\psi(t_{c,n})) \in \mathbb{R}^D$ with feature dimension $D$, forming the aggregate attribute matrix $\mathbf{T}^{can} \in \mathbb{R}^{C \times N \times D}$.

Using all $N$ attributes introduces redundancy and computational overhead. We greedily select a compact subset of $L$ attributes ($L < N$) via marginal gain, balancing two objectives. *Representativeness* measures an attribute's thematic relevance to others, reflecting its capacity to capture common cross-class patterns. *Diversity* quantifies an attribute's capacity to introduce complementary semantics beyond already selected ones. Jointly optimizing both objectives ensures the selected subset is both semantically coherent

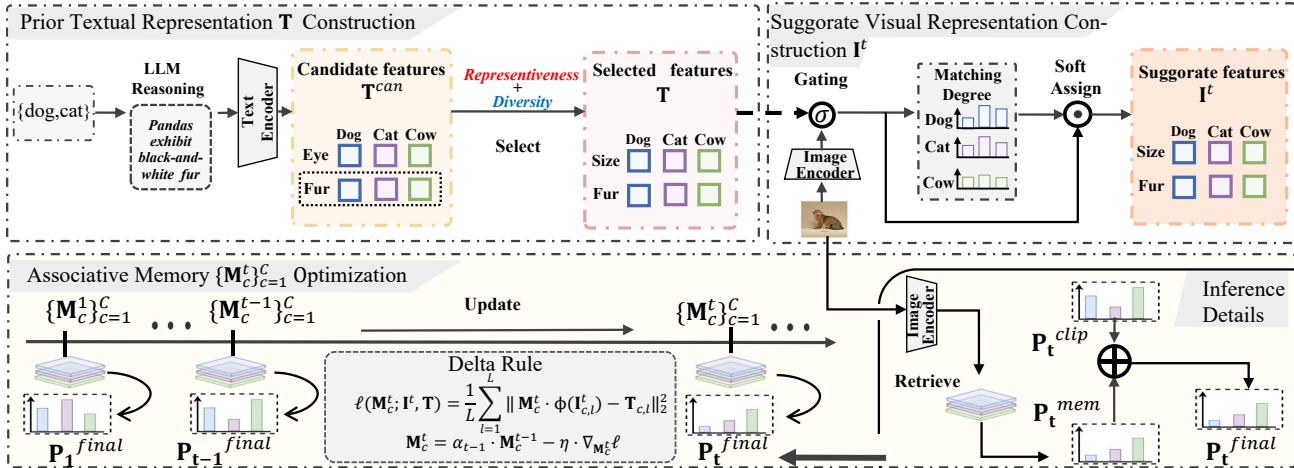

*Figure 2.* **Overview of the A$^2$Memory framework.** The framework builds attribute-level textual representations for semantic guidance, performs attribute-guided soft assignment for incoming samples, and updates class-wise associative memories for test-time adaptation.

and maximally informative. For a candidate index $k$, we define:

$$\text{Rep}(k) = \frac{1}{C|\mathcal{S}_{can}|} \sum_{c=1}^{C} \sum_{i \in \mathcal{S}_{can} \setminus k} \langle \mathbf{T}_{c,k}^{can}, \mathbf{T}_{c,i}^{can} \rangle, \quad (5)$$

$$\text{Div}(k) = 1 - \frac{1}{C|\mathcal{S}_{sel}|} \sum_{c=1}^{C} \sum_{j \in \mathcal{S}_{sel}} \langle \mathbf{T}_{c,k}^{can}, \mathbf{T}_{c,j}^{can} \rangle,$$

where $\mathcal{S}_{can} = \{1, \dots, N\}$ is the initial set of candidate attributes and $\mathcal{S}_{sel} = \emptyset$ is the initial empty set of selected attributes. We start by selecting the most representative attribute $k^* = \arg\max_k \text{Rep}(k)$, then iteratively choose attributes based on marginal gain with balancing factor $\lambda$:

$$k^* = \arg\max_{k \in S_{can}} [\lambda \cdot \text{Rep}(k) + (1-\lambda) \cdot \text{Div}(k)]. \quad (6)$$

After selection, the attribute is transferred from $S_{can}$ to $S_{sel}$:

$$\mathcal{S}_{sel} \leftarrow \mathcal{S}_{sel} \cup \{k^*\}, \quad \mathcal{S}_{can} \leftarrow \mathcal{S}_{can} \setminus \{k^*\}. \quad (7)$$

Until $|\mathcal{S}_{sel}| = L$, the resulting prior textual representation $\mathbf{T} \in \mathbb{R}^{C \times L \times D}$ is obtained by selecting from the candidate representations $\mathbf{T}^{can}$, where each entry is defined as:

$$\mathbf{T}_{c,l} = \mathbf{T}_{c,\pi(l)}^{can}, \quad \pi(l) \in \mathcal{S}_{sel}. \quad (8)$$

Here, $\pi(\cdot)$ denotes an indexing function that maps the $l$-th selected attribute to its corresponding index in the candidate set. The resulting prior textual representation $\mathbf{T} \in \mathbb{R}^{C \times L \times D}$ organizes $L$ representative and diverse attributes shared across all $C$ classes. Beyond offline construction, this shared attribute structure enables attribute-level guidance for incoming test samples, aligning visual features with specific attribute slots across all classes.

**Attribute-Centric Soft Assignment.** Existing memory-based methods (Zhang et al., 2024a; Karmanov et al., 2024) assign each test sample to a single class memory based on CLIP's pseudo-label. This hard assignment fails when CLIP's prediction is incorrect, leading to memory contamination . Leveraging our shared attribute structure, we apply attribute-centric soft assignment that distributes each test sample across all class-attribute slots simultaneously, down-weighting low-confidence classes while preserving informative signals from probable candidates.

Given test image $\boldsymbol{x}_t$ with feature $\boldsymbol{f}^t = \text{norm}(f_\theta(\boldsymbol{x}_t)) \in \mathbb{R}^D$, we compute surrogate features through attribute-wise alignment. For each class-attribute pair $(c, l)$:

$$\mathbf{I}_{c,l}^t = \text{norm}(\boldsymbol{f}^t + \text{norm}(\boldsymbol{f}^t \odot \sigma(\boldsymbol{f}^t \odot \mathbf{T}_{c,l}))), \quad (9)$$

where $\odot$ denotes element-wise product and $\sigma(\cdot)$ denotes the sigmoid function. The inner term $(\boldsymbol{f}^t \odot \mathbf{T}_{c,l})$ computes dimension-wise visual-attribute alignment, and the sigmoid function applies textual-guided gating that highlights consistent dimensions while suppressing conflicting ones. The residual connection preserves original visual information. The resulting *Surrogate Visual Representation* $\mathbf{I}^t \in \mathbb{R}^{C \times L \times D}$ organizes visual features in a class-wise manner with attribute-centric soft alignment, where for each class $c$, each entry $\mathbf{I}_{c,l}^t$ encodes the visual evidence aligned to the $l$-th attribute, allowing confident samples to concentrate more relevant attributes while ambiguous ones distribute their responses more evenly across the attribute dimension.

### 3.3. Class-wise Associative Memory Optimization

Existing memory-based methods (Karmanov et al., 2024; Zhang et al., 2025) maintain a fixed-capacity memory that cannot retain all incoming samples. When capacity is exceeded, potentially informative samples are inevitably

discarded, which limits memory expressiveness. Drawing on recent advances in associative memory (Behrouz et al., 2025a;b), we design class-wise associative memories that dynamically compress streaming samples into fixed-capacity memory through gradient-based optimization, preserving discriminative information from historical samples.

**Memory Architecture and Formulation.** For each class $c$, we maintain an associative memory matrix $\mathbf{M}_c^t \in \mathbb{R}^{D \times D}$ that is learned at step $t$ and maps surrogate visual representation $\mathbf{I}^t$ as keys to prior textual representations $\mathbf{T}$ as values. The memory is dynamically updated based on streaming key-value pairs, with retrieval performed using the current sample feature $\boldsymbol{f}^t$ as the query, through $\mathbf{M}_c^t \cdot \phi(\boldsymbol{f}^t)$, where $\phi(\mathbf{x}) = \frac{\mathrm{SiLU}(\mathbf{x})}{\|\mathrm{SiLU}(\mathbf{x})\|_2}$ and $\mathrm{SiLU}(\mathbf{x}) = \mathbf{x} \odot \sigma(\mathbf{x})$ (Elfwing et al., 2018). We formulate memory learning based on previous approaches (Yang et al., 2024a). For class $c$, we define the $\ell_2$ loss function aggregated across the $L$-attribute batch:

$$\ell(\mathbf{M}_c^t; \mathbf{I}^t, \mathbf{T}) = \frac{1}{L} \sum_{l=1}^{L} \big\| \mathbf{M}_c^t \cdot \phi(\mathbf{I}_{c,l}^t) - \mathbf{T}_{c,l} \big\|_2^2. \quad (10)$$

The memory updates more strongly for samples that violate its current expectations. The batch averaging over $L$ attributes aggregates update signals from multiple attribute perspectives, providing implicit regularization that attenuates the impact of noisy individual attributes.

**Optimization and Retention.** Optimizing the loss via gradient descent yields the delta rule update. The gradient with respect to the memory matrix $\mathbf{M}_c^t$ is:

$$\nabla_{\mathbf{M}_c^t} \ell = \frac{2}{L} \sum_{l=1}^{L} \big( \mathbf{M}_c^t \cdot \phi(\mathbf{I}_{c,l}^t) - \mathbf{T}_{c,l} \big) \phi(\mathbf{I}_{c,l}^t)^\top. \quad (11)$$

The resulting associative memory is updated as:

$$\mathbf{M}_c^t = \alpha_{t-1} \cdot \mathbf{M}_c^{t-1} - \eta \cdot \nabla_{\mathbf{M}_c^t} \ell, \quad (12)$$

where $\eta$ is the learning rate and $\alpha_{t-1}$ is a data-dependent retention coefficient that governs the plasticity-stability trade-off and is updated over time based on the confidence of the final prediction $\boldsymbol{P}_t^{final}$ of the current sample in Eq. (15):

$$\alpha^t = \frac{t-1}{t} \alpha^{t-1} + \frac{1}{t}\left(1 - \tilde{H}\left(\boldsymbol{P}_t^{final}\right)\right), \quad (13)$$

where $\tilde{H}\left(\boldsymbol{P}_t^{final}\right) = -\frac{1}{\log C} \sum_{c=1}^{C} p_c^{final} \log p_c^{final}$ is the normalized prediction entropy used to measure confidence. A formal convergence analysis of this gradient-based update rule is provided in Appendix A.

### 3.4. The Workflow of $A^2$Memory in Prediction

In this section, we introduce the inference procedure of $A^2$Memory, which performs memory-based retrieval and confidence-aware fusion to generate final predictions. The procedure is detailed in Algorithm 1. Given optimized class-wise associative memories $\{\mathbf{M}_c^t\}_{c=1}^C$, we generate predictions through memory retrieval and adaptive branch fusion. For query $\boldsymbol{q}^t = \phi(\boldsymbol{f}^t)$, we retrieve class information by mapping through each associative memory:

$$\boldsymbol{o}_c^t = \boldsymbol{q}^{t\top}\mathbf{M}_c^t, \quad p_c^{mem} = \langle \boldsymbol{o}_c^t, \boldsymbol{f}^t \rangle. \quad (14)$$

The term $\boldsymbol{o}_c^t \in \mathbb{R}^D$ denotes the sample-adaptive class prototype retrieved from the associative memory, which is dynamically reconstructed to optimally correspond to the current query sample. The subsequent inner product with $\boldsymbol{f}^t$ quantifies the alignment between the original visual feature and this adaptive reference. This formulation enables each class memory to produce an instance-specific classification standard, enriching the discriminative signal beyond rigid feature-prototype comparisons.

We then compute CLIP's zero-shot logits using attribute-averaged text features as $p_c^{clip} = \left\langle \boldsymbol{f}^t, \frac{1}{L} \sum_{l=1}^{L} \mathbf{T}_{c,l} \right\rangle$. The averaging over $L$ attributes provides a more diverse class representation than single-template embeddings. The two prediction branches capture complementary information where zero-shot predictions leverage CLIP's pre-trained knowledge, and memory-based predictions incorporate historical data statistics. We stack the class-wise logits from each branch to form the two vectors $\boldsymbol{P}_t^{clip} = [p_1^{clip}, \ldots, p_C^{clip}]^\top$ and $\boldsymbol{P}_t^{mem} = [p_1^{mem}, \ldots, p_C^{mem}]^\top$, which are then fused via confidence-weighted combination by:

$$\boldsymbol{P}_t^{final} = \lambda^{clip} \cdot \boldsymbol{P}_t^{clip} + \lambda^{mem} \cdot \boldsymbol{P}_t^{mem}, \quad (15)$$

where $\lambda^{clip} = 1 - \tilde{H}\left(\boldsymbol{P}_t^{clip}\right)$ and $\lambda^{mem} = 1 - \tilde{H}(\boldsymbol{P}_t^{mem})$. This sample-adaptive scheme assigns higher weight to the more confident branch, leading to more reliable inference.

## 4. Related Work

**Vision-Language Test-Time Adaptation.** Vision-language models (VLMs), pretrained on large-scale image–text pairs via contrastive learning (Radford et al., 2021; Zhai et al., 2023; Siméoni et al., 2025), demonstrate strong performance across diverse downstream tasks. However, models like CLIP (Radford et al., 2021) are sensitive to distribution shifts (Zhou et al., 2022b;a), which motivates test-time adaptation (TTA) that adapts models to test distribution. Test-time prompt tuning (Shu et al., 2022; Yoon et al., 2024; Xiao et al., 2025) has emerged to optimize learnable prompts via entropy minimization. Despite their effectiveness, these methods incur computational overhead due to iterative optimization. To address this efficiency bottleneck, training-free TTA methods have gained increasing attention. Distribution-level methods model target-domain characteristics through

*Table 1.* Top-1 accuracy (%) comparison on Cross Datasets. The best results are **bolded** and the second-best results are underlined.

| Method | Aircraft | Caltech | Cars | DTD | EuroSAT | Flower | Food101 | Pets | SUN397 | UCF101 | **Avg.** |
|---|---|---|---|---|---|---|---|---|---|---|---|
| CLIP-RN50 (Radford et al., 2021) | 16.11 | 87.26 | 55.89 | 40.37 | 25.79 | 62.77 | 74.82 | 82.97 | 60.85 | 59.48 | 56.63 |
| TPT (Shu et al., 2022) | 17.58 | 87.02 | 58.46 | 40.84 | 28.33 | 62.69 | 74.88 | 84.49 | 61.46 | 60.82 | 57.66 |
| DPE (Zhang et al., 2024a) | 19.80 | **90.83** | 59.26 | 50.18 | 41.67 | 67.60 | 77.83 | 85.97 | 64.23 | 61.98 | 61.94 |
| Dota (Han et al., 2024) | 18.06 | 88.84 | 58.72 | 45.80 | 47.15 | 68.53 | **78.61** | **87.33** | 63.89 | **65.08** | 62.20 |
| BCA (Zhou et al., 2025) | 19.89 | 89.70 | 58.13 | 48.58 | 42.12 | 66.30 | 77.19 | 85.58 | 63.38 | 63.51 | 61.44 |
| TDA (Karmanov et al., 2024) | 17.61 | 89.70 | 57.78 | 43.74 | 42.11 | 68.74 | 77.75 | 86.18 | 62.53 | 64.18 | 61.03 |
| ADAPT (Zhang et al., 2025) | 18.00 | 89.37 | 58.38 | **51.89** | 50.47 | **70.04** | 75.57 | 86.43 | 63.12 | 64.29 | 62.82 |
| **Ours** | **19.91** | 90.36 | **59.66** | **51.89** | **51.22** | 69.91 | 76.26 | 86.92 | **65.22** | 64.29 | **63.56** |
| CLIP-ViT/16 (Radford et al., 2021) | 23.22 | 93.55 | 66.11 | 45.04 | 50.42 | 66.99 | 82.86 | 86.92 | 65.63 | 65.16 | 64.59 |
| TPT (Shu et al., 2022) | 24.78 | 94.16 | 66.87 | 47.75 | 42.44 | 68.98 | 84.67 | 87.79 | 65.50 | 68.04 | 65.10 |
| DPE (Zhang et al., 2024a) | 28.95 | 94.81 | 67.31 | 54.20 | 55.79 | 75.07 | 86.17 | 91.14 | 70.07 | 70.44 | 69.40 |
| TDA (Karmanov et al., 2024) | 23.91 | 94.24 | 67.28 | 47.40 | 58.00 | 71.42 | 86.14 | 88.63 | 67.62 | 70.66 | 67.53 |
| DynaProm (Xiao et al., 2025) | 24.33 | 94.32 | 67.65 | 47.96 | 42.28 | 69.95 | 85.42 | 88.28 | 66.32 | 68.72 | 65.52 |
| BCA (Zhou et al., 2025) | 28.59 | 94.69 | 66.86 | 53.49 | 56.63 | 73.12 | 85.97 | 90.43 | 68.41 | 67.59 | 68.58 |
| TCA (Wang et al., 2025) | 24.87 | 93.63 | 65.33 | 46.16 | **70.43** | 73.33 | 85.31 | 89.53 | 65.92 | **72.38** | 68.69 |
| Dota (Han et al., 2024) | 25.59 | 94.32 | **69.48** | 47.87 | 57.65 | 74.67 | **87.02** | 91.69 | 69.70 | 72.06 | 69.01 |
| ADAPT (Zhang et al., 2025) | 28.95 | 94.48 | 68.19 | 55.20 | 68.19 | 75.56 | 83.81 | 92.01 | **70.57** | 70.66 | 70.76 |
| **Ours** | **29.92** | **95.01** | 68.20 | **57.28** | 68.19 | **76.11** | 86.19 | **92.14** | 70.43 | 70.66 | **71.41** |

Gaussian assumptions (Wang et al., 2024; Han et al., 2024; Dai & Yang, 2025; Zanella et al., 2025), while optimal transport methods (Zhu et al., 2024; Chen et al., 2025a) formulate adaptation as cross-modal distribution alignment. Recently, memory-based methods have emerged as the dominant paradigm due to their lightweight computation and competitive performance. TDA (Karmanov et al., 2024) introduces dual memories to store high-confidence features, while DPE (Zhang et al., 2024a) maintains dual-modal class prototypes with residual parameters to align multi-modal memory representations. Subsequent works further improve memory construction (Zhang et al., 2024b; Wang et al., 2025; Zhang et al., 2024d) and memory utilization (Huang et al., 2025; Zhang et al., 2025; Guan et al., 2026). However, these methods rely on hard pseudo-label assignment and memory eviction, causing memory contamination and limiting memory expressiveness.

**Associative Memory.** Attention (Vaswani et al., 2017) computes outputs by matching queries against keys to retrieve corresponding values, a process that can be interpreted through the lens of associative memory (Hopfield, 1982; Ramsauer et al., 2020). Associative memory (Hopfield, 1982; Krotov & Hopfield, 2016) is an operator mapping keys as addressable patterns to values as retrievable contents. Recent work reinterprets sequence models as test-time memorization modules that compress streaming inputs into fixed-capacity parametric memory, learning new key-value associations while retaining previously memorized information (Behrouz et al., 2025a;b). Specifically, Hebbian-like rules (Hebb, 2005) accumulate key-value associations (Yang et al., 2023; Sun et al., 2023) while the Delta rule (Rajpurkar et al., 2016) updates memory via gradient descent based on attentional bias (Behrouz et al., 2025b). To prevent catastrophic forgetting, retention regularization (Behrouz et al., 2025b) through data-dependent gating (Yang et al., 2023; Zhang et al., 2024c) selectively

preserves past memory for balancing plasticity and stability. Our work first bridges these advances to vision-language TTA, reformulating memory-based adaptation as associative memory optimization.

## 5. Experiments

### 5.1. Experimental Setup.

**Datasets.** Following prior works (Karmanov et al., 2024; Zhang et al., 2024a), we evaluate on two benchmarks: Cross-Dataset Generalization (CD) and Robustness to Out-of-Distribution Shifts (OOD). The former comprises 10 diverse datasets: Aircraft (Maji et al., 2013), Caltech101 (Fei-Fei, 2004), Cars (Krause et al., 2013), Describable Textures (DTD) (Cimpoi et al., 2014), EuroSAT (Helber et al., 2019), Flowers102 (Nilsback & Zisserman, 2008), Food101 (Bossard et al., 2014), Pets (Parkhi et al., 2012), SUN397 (Sun et al., 2023), and UCF101 (Soomro et al., 2012). The latter consists of ImageNet (Deng et al., 2009), ImageNetV2 (Recht et al., 2019), ImageNet-R (Hendrycks et al., 2021a), ImageNet-Sketch (Wang et al., 2019), and ImageNet-A (Hendrycks et al., 2021b).

**Implementation Details.** We adopt the CLIP-ViT-B/16 and CLIP-RN50 (Radford et al., 2021) as the backbone. We conduct evaluations in a fully online, streaming manner with a batch size of 1. Regarding hyperparameters, we specify the number of candidate textual descriptions for each class as $N = 15$, from which $L = 5$ final visual features are selected by maximizing marginal gain, balanced by $\lambda = 0.3$. The hyperparameter $\eta$ is set to 0.01. All experiments are conducted on one NVIDIA A100 GPU. We report top-1 accuracy following the TDA (Karmanov et al., 2024) protocol.

**Baselines.** We compare our method against zero-shot CLIP (Radford et al., 2021) and recent TTA methods:

(i) *Optimization-based methods* that require backpropagation, including TPT (Shu et al., 2022), DPE (Zhang et al., 2024a) and DynaPrompt (Xiao et al., 2025); and (ii) *Backpropagation-free approaches* that adapt via statistics estimation or memory, including Dota (Han et al., 2024), BCA (Zhou et al., 2025), TDA (Karmanov et al., 2024), TCA (Wang et al., 2025), and ADAPT (Zhang et al., 2025).

## 5.2. Main Results

**Evaluation of Cross-Datasets Generalization.** As shown in Table 1, our method achieves the highest average accuracy of **71.41%** with CLIP-ViT-B/16, exceeding zero-shot CLIP by 6.83% and outperforming all prior training-free and optimization-based TTA methods. Notable gains are observed on DTD (**57.28%**, +2.08% over ADAPT) and Food101 (**86.19%**, +2.38%). Similar improvements are obtained with the ResNet-50 backbone, where our method reaches **63.56%** average accuracy, confirming robust generalization across architectures and domains.

**Evaluation of Natural Distribution Shifts.** We further assess robustness under natural distribution shifts, as shown in Table 2. With the ViT-B/16 backbone, our method achieves the highest average accuracy of **67.23%**, consistently outperforming all baselines, demonstrating exceptional resilience on ImageNet-A (**64.38%**) and ImageNet (**72.21%**), which contain severe distribution shifts. Similarly, with the ResNet-50, our method maintains superiority with an average accuracy of **51.70%** and an OOD average of **48.63%**. These results highlight that our method effectively mitigates domain shifts in an online manner, offering a robust solution for real-world test-time adaptation.

## 5.3. Analysis and Ablations

**Component Analysis.** We ablate key components in $A^2$Memory to validate their contributions (Table 3). First, removing prior textual representation $\mathbf{T}$ degrades performance, verifying its role in establishing diverse, attribute-centric guidance of visual semantics. Moreover, discarding surrogate visual representation $\mathbf{I^t}$ from attribute-centric soft assignment leads to consistent drops, confirming that distributing samples at the feature level across attributes effectively mitigates memory contamination caused by hard pseudo-labels. Finally, removing class-wise associative memory $\{\mathbf{M}_c^t\}_{c=1}^C$ causes significant degradation, demonstrating that gradient-based compression significantly enhances memory expressiveness by capturing rich historical information within the attribute subspace, rather than discarding it.

**Impact of Data-Dependent Retention coefficient $\alpha_t$.** To validate our entropy-weighted adaptive retention, Table 4 compares our method against data-independent baselines, including fixed coefficients ($\alpha_t \in \{0.1, 0.5, 0.9\}$) and

*Table 2.* Top-1 accuracy (%) comparison on OOD datasets.

| Method | Image Net | IN -A | IN -V | IN -R | IN -S | **OOD** Avg. | Avg. |
|---|---|---|---|---|---|---|---|
| CLIP-RN50 | 59.81 | 23.24 | 52.91 | 60.72 | 35.48 | 43.09 | 46.43 |
| TPT | 60.74 | 26.67 | 54.70 | 59.11 | 35.09 | 43.89 | 47.26 |
| TDA | 61.35 | 30.29 | 55.54 | 62.58 | 38.12 | 46.63 | 49.58 |
| DPE | 63.41 | 30.15 | **56.72** | 63.72 | 40.03 | 47.66 | 50.81 |
| DynaPrompt | 61.56 | 27.84 | 55.12 | 60.63 | 35.64 | 44.81 | 48.16 |
| BCA | 61.81 | 30.35 | 56.58 | 62.89 | 38.04 | 46.96 | 49.93 |
| ADAPT | 62.16 | 33.08 | 55.97 | 62.69 | **40.21** | 47.99 | 50.82 |
| **Ours** | **63.96** | **34.51** | 55.99 | **63.86** | 40.16 | **48.63** | **51.70** |
| CLIP-ViT/16 | 68.34 | 49.89 | 61.88 | 77.65 | 48.24 | 59.42 | 61.20 |
| TPT | 68.98 | 54.77 | 63.45 | 77.06 | 47.97 | 60.81 | 62.45 |
| DPE | 71.91 | 59.63 | **65.44** | 80.40 | 52.26 | 64.43 | 65.93 |
| TDA | 69.51 | 60.11 | 64.67 | 80.24 | 50.54 | 63.89 | 65.01 |
| DynaPrompt | 69.61 | 56.17 | 64.67 | 78.17 | 48.22 | 61.81 | 63.37 |
| BCA | 70.22 | 61.14 | 64.90 | 80.72 | 50.87 | 64.41 | 65.57 |
| TCA | 68.88 | 50.13 | 62.10 | 77.11 | 48.95 | 59.57 | 61.43 |
| ADAPT | 70.91 | 63.32 | 64.64 | 80.66 | 53.13 | 65.44 | 66.53 |
| **Ours** | **72.21** | **64.38** | 65.38 | **80.98** | 53.21 | **65.99** | **67.23** |

*Table 3.* **Component analysis.** We evaluate the impact of removing $\mathbf{T}$, $\mathbf{I^t}$, and $\{\mathbf{M}_c^t\}_{c=1}^C$. The first row represents zero-shot clip.

| Components | | | ViT-B/16 | | RN50 | |
|---|---|---|---|---|---|---|
| $\mathbf{T}$ | $\mathbf{I^t}$ | $\{\mathbf{M}_c^t\}_{c=1}^C$ | OOD | CD | OOD | CD |
| ✗ | ✗ | ✗ | 59.42 | 64.59 | 43.09 | 56.63 |
| ✓ | ✓ | ✗ | 63.45 | 68.95 | 46.88 | 61.05 |
| ✓ | ✗ | ✓ | 64.12 | 69.45 | 47.50 | 61.95 |
| ✗ | ✓ | ✓ | 63.80 | 68.50 | 46.20 | 60.80 |
| ✓ | ✓ | ✓ | **65.99** | **71.41** | **48.63** | **63.56** |

heuristic updates (random, cumulative average). Our adaptive strategy consistently outperforms all approaches (e.g., +1.86% on ViT-B/16 OOD by $\alpha = 0.9$). Random updates destabilize convergence and cumulative averages dilute discriminative signals, while fixed strategies are trapped between excessive inertia retaining redundant history (large $\alpha_t$) and catastrophic forgetting (small $\alpha_t$). In contrast to these static rules, our approach introduces a data-dependent mechanism that dynamically modulates the plasticity-stability trade-off based on instance confidence.

**Efficiency Analysis.** Table 5 evaluates the computational cost and performance on ImageNet with ViT-B/16. Compared to optimization-based methods, our approach demonstrates superior efficiency by eliminating expensive backpropagation. Specifically, it achieves a 8.4× speedup over TPT and is 1.6× faster than DPE. Among BP-free baselines, our method outperforms the ADAPT in both inference speed and accuracy. While TDA is marginally faster due to

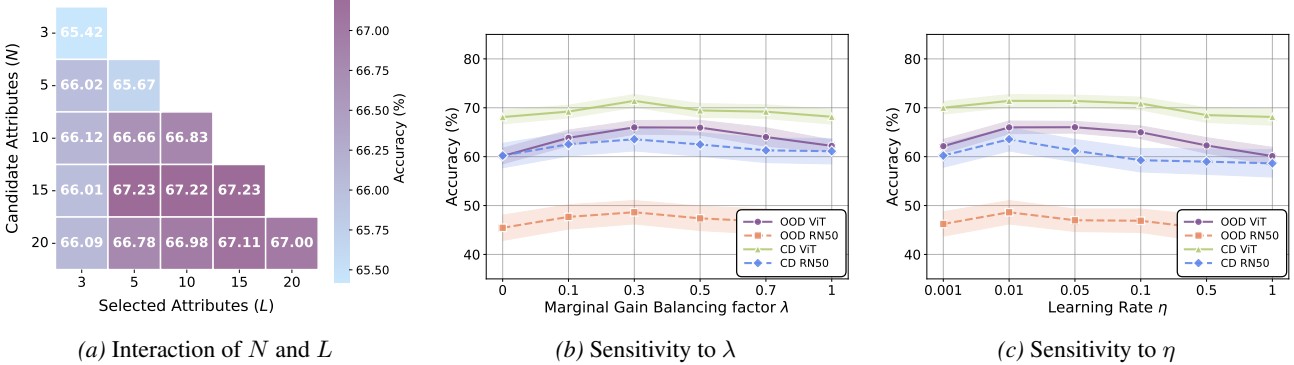

*(a)* Interaction of $N$ and $L$          *(b)* Sensitivity to $\lambda$          *(c)* Sensitivity to $\eta$

*Figure 3.* **Hyperparameter Sensitivity Analysis.** (a) Accuracy heatmap w.r.t candidate size $N$ and selection size $L$. (b) Performance stability across different marginal gain balancing factors $\lambda$. (c) Impact of learning rate $\eta$ on memory adaptation performance.

*Table 4.* **Analysis of memory update strategies.** Entropy-weighted adaptive retention coefficient ($\alpha_t$) against fixed values, random updates, and cumulative average.

| Update Strategy ($\alpha_t$) | ViT-B/16 | | RN50 | |
|---|---|---|---|---|
| | OOD | CD | OOD | CD |
| Random Update | 41.55 | 36.40 | 34.15 | 48.20 |
| Cumulative Average | 59.90 | 64.15 | 45.80 | 59.40 |
| Fixed $\alpha_t = 0.9$ | 64.23 | 69.35 | 47.10 | 62.90 |
| Fixed $\alpha_t = 0.5$ | 62.50 | 68.80 | 47.45 | 62.15 |
| Fixed $\alpha_t = 0.1$ | 62.80 | 68.10 | 46.20 | 60.50 |
| **Adaptive (Ours)** | **65.99** | **71.41** | **48.63** | **63.56** |

*Table 5.* **Efficiency Comparison.** Efficiency on Image-Net with ViT-B/16. "BP-free" denotes adaptation without backpropagation.

| Method | BP-free | Time(min) | Acc (%) | Gain (%) |
|---|---|---|---|---|
| CLIP | ✓ | 10.01 | 68.34 | - |
| TPT | ✗ | 586.43 | 68.98 | +0.64 |
| DPE | ✗ | 114.23 | 71.91 | +3.45 |
| TDA | ✓ | **52.50** | 69.51 | +1.17 |
| ADAPT | ✓ | 78.55 | 70.91 | +2.57 |
| **Ours** | ✓ | 69.24 | **72.21** | **+3.87** |

*Table 6.* **Memory branch analysis** on ViT-B/16. $\boldsymbol{P}^{mem}$-only uses the associative memory branch without CLIP fusion.

| Method | ImageNet | OOD Avg | CD Avg |
|---|---|---|---|
| TDA | 69.51 | 63.89 | 67.53 |
| ADAPT | 70.91 | 65.44 | 70.76 |
| $\boldsymbol{P}^{mem}$-only | 71.34 | 65.83 | 71.26 |
| **A²Memory** | **72.21** | **65.99** | **71.41** |

itself, rather than from the fusion strategy. Fusing with CLIP's zero-shot branch via confidence-weighted combination further boosts performance, as the two branches capture complementary information: zero-shot predictions leverage CLIP's pre-trained knowledge while memory-based predictions incorporate target domain statistics accumulated over the stream.

**Hyperparameter Sensitivity Analysis.** Regarding attribute memory construction (Fig. 3(a)), performance generally improves as the number of candidate ($N$) and selected ($L$) attributes increases, indicating that a richer semantic vocabulary captures finer-grained visual details. However, accuracy saturates around $N = 20$, suggesting that a compact attribute set is sufficient to cover the visual-semantic space. Second, for the marginal gain selection (Fig. 3(b)), the model achieves optimal performance when $\lambda = 0.3$, confirming the necessity of balancing representativeness and diversity. For memory adaptation learning rate $\eta$ in Fig. 3(c), performance is best around 0.01, whereas large rates destabilize memory, and small rates hinder adaptation.

its simplistic memory design, it suffers from a significant performance drop (-2.7% compared to ours). These results highlight our method as a practical solution balancing accuracy and efficiency.

**Memory Branch Analysis.** To understand the contribution of the associative memory branch, we evaluate $\boldsymbol{P}^{mem}$-*only*, which uses the memory branch prediction alone without CLIP zero-shot fusion (ViT-B/16). As shown in Table 6, the memory branch alone already surpasses ADAPT on all metrics, demonstrating that gradient-based associative memory produces strong and reliable adaptation signals even in isolation. This confirms that the performance gains of A²Memory stem primarily from the memory mechanism

**In-Depth Analysis.** In Figure 4, we provide an in-depth analysis of our method in addressing two challenges. For challenge ❶, as shown in Fig. 4(left), our method achieves significantly higher memory accuracy in the early stages, outperforming CLIP and others, consistently surpassing baselines in later stages. This also demonstrates that our

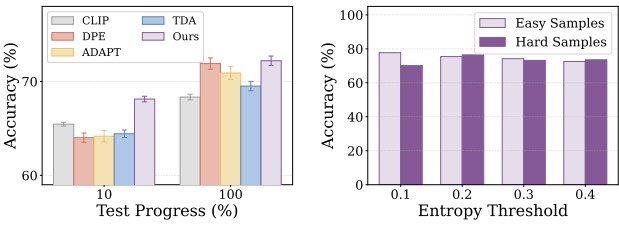

*Figure 4.* In-depth analysis of $A^2$Memory for two challenges.

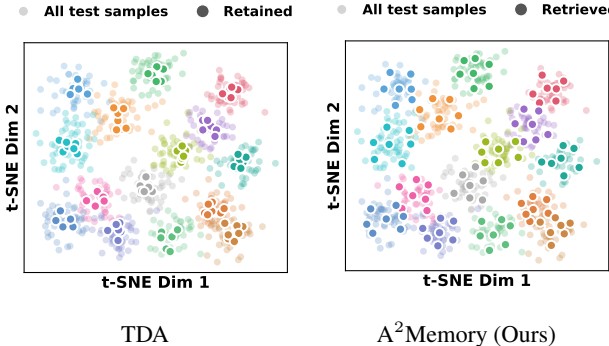

*Figure 5.* t-SNE visualization of retained / retrieved features on ImageNet (ViT-B/16). Dark dots indicate retained samples; colors denote classes. TDA's entropy-based eviction concentrates retained samples near cluster cores (left), while $A^2$Memory achieves uniform coverage across the full test distribution (right).

memory mechanism is effective from the start, which is not observed in previous memory-based methods. Unlike prior methods which suffer from memory contamination due to hard assignments, our method leverages attribute-guided soft assignment, effectively alleviating error accumulation. For challenge ❷, as shown in Fig. 4(right), our method maintains balanced accuracy for both easy and hard samples across varying entropy thresholds. This demonstrates that samples contribute equally to the memory, alleviating bias towards easier samples in memory. By leveraging associative memory, our method compresses diverse and valuable information for effective adaptation through attribute-level compression.

**Feature Coverage Visualization.** Figure 5 presents t-SNE visualizations comparing the feature distributions retained by TDA and $A^2$Memory on ImageNet (ViT-B/16). In TDA, entropy-based eviction causes retained samples (dark dots) to collapse into narrow, high-confidence clusters near the cluster core, severely limiting coverage of the test distribution. In contrast, $A^2$Memory's gradient-based associative memory compresses information from all observed samples without hard eviction, resulting in retrieving prototypes that spread uniformly across each class cluster. This broader coverage enables more accurate and robust retrieval during inference, directly explaining the performance gains on challenging and under-represented samples.

# 6. Conclusion

We identify two key challenges of memory-based VLM test-time adaptation: memory contamination from hard pseudo-label assignment, which biases subsequent predictions, and biased selection under fixed memory capacity, which limits its memory effectiveness. To address these, we propose $A^2$Memory. It constructs attribute-level prior textual representations as semantic guidance and adopts attribute-centric soft assignment to generate surrogate visual representations, thereby alleviating memory contamination by avoiding hard assignment. In addition, $A^2$Memory parameterizes memory as class-wise associative mappings and updates them through gradient-based optimization with entropy-adaptive retention to retrieve sample-adaptive class prototypes, preserving discriminative information from all observed samples. Experimental results show that $A^2$Memory consistently enhances test-time adaptation performance across 15 benchmarks.

# Impact Statement

This paper presents $A^2$Memory, a novel framework for Test-Time Adaptation (TTA) of Vision-Language Models. This advancement contributes to the reliability and safety of AI systems deployed in dynamic real-world environments, such as autonomous driving and robotics. We do not foresee immediate negative societal consequences from this work, although standard ethical considerations regarding the general application of visual recognition systems apply.

# Acknowledgments

Z. Lin is supported by the Beijing Major Science and Technology Project (No. Z251100008425006), the Beijing Natural Science Foundation (No. L257007), the NSF China (No. 62276004), and the State Key Laboratory of General Artificial Intelligence. H. Li is supported by the Beijing Major Science and Technology Project (No. Z251100008425006) and the Beijing Natural Science Foundation (No. L257007). Y. Wang is supported by the Beijing Major Science and Technology Project (No. Z251100008425006), the NSF China (Nos. 92370129 and 62376010), the Beijing Natural Science Foundation (No. L257007), and the Beijing Nova Program (Nos. 20230484344 and 20240484642).

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

## Appendix

## A. Convergence Analysis of Class-wise Associative Memory Optimization

We provide a formal convergence guarantee for the gradient-based memory update in Eq. (12).

**Setup.** Let $\phi_l := \phi(\mathbf{I}_{c,l}^t) \in \mathbb{R}^D$, $\|\phi_l\|_2 = 1$, $\mathbf{t}_l := \mathbf{T}_{c,l} \in \mathbb{R}^D$. Eq. (12) optimizes $\tilde{\ell}(\mathbf{M}_c) = \ell(\mathbf{M}_c) + \frac{1-\alpha_{t-1}}{2\eta}\|\mathbf{M}_c\|_F^2$. Let $\mathbf{M}_c^* = \arg\min \tilde{\ell}$, $\varepsilon_0 = \|\mathbf{M}_c^{(0)} - \mathbf{M}_c^*\|_F$, $\alpha_{t-1} \in (0,1)$, $\eta > 0$.

1. **Smoothness.** $\nabla\tilde{\ell}$ is $\tilde{L}$-Lipschitz, $\tilde{L} = 2 + \frac{1-\alpha_{t-1}}{\eta}$. For any $\mathbf{A}, \mathbf{B}$: $\|\nabla\tilde{\ell}(\mathbf{A}) - \nabla\tilde{\ell}(\mathbf{B})\|_F \leq \left(2 + \frac{1-\alpha_{t-1}}{\eta}\right)\|\mathbf{A} - \mathbf{B}\|_F$.

2. **Initialization.** $\|\mathbf{M}_c^{(0)} - \mathbf{M}_c^*\|_F = \varepsilon_0 < \infty$.

**Theorem A.1** (Convergence Rate). *Using conditions above, for all $K > 0$: $\tilde{\ell}\left(\mathbf{M}_c^{(K)}\right) - \tilde{\ell}(\mathbf{M}_c^*) \leq \frac{\varepsilon_0^2}{2\eta K}$.*

*Proof.* **Convexity.** At update time $t$, $\phi_l$ and $\mathbf{t}_l$ are fixed. Thus, $\|\mathbf{M}_c\phi_l - \mathbf{t}_l\|_2^2$ is the composition of an affine map with a squared norm, making $\tilde{\ell}$ a **convex quadratic** in $\mathbf{M}_c$.

Since $\tilde{\ell}$ is convex and $\tilde{L}$-smooth, applying the descent lemma to $\mathbf{M}_c^{(k+1)} = \mathbf{M}_c^{(k)} - \eta\nabla\tilde{\ell}(\mathbf{M}_c^{(k)})$ and setting $\eta \leq 1/\tilde{L}$:

$$\tilde{\ell}(\mathbf{M}_c^{(k+1)}) \leq \tilde{\ell}(\mathbf{M}_c^{(k)}) + \langle\nabla\tilde{\ell}(\mathbf{M}_c^{(k)}), \mathbf{M}_c^{(k+1)} - \mathbf{M}_c^{(k)}\rangle_F + \frac{\tilde{L}}{2}\|\mathbf{M}_c^{(k+1)} - \mathbf{M}_c^{(k)}\|_F^2$$

$$= \tilde{\ell}(\mathbf{M}_c^{(k)}) - \eta\|\nabla\tilde{\ell}(\mathbf{M}_c^{(k)})\|_F^2 + \frac{\tilde{L}\eta^2}{2}\|\nabla\tilde{\ell}(\mathbf{M}_c^{(k)})\|_F^2$$

$$= \tilde{\ell}(\mathbf{M}_c^{(k)}) - \eta\left(1 - \frac{\tilde{L}\eta}{2}\right)\|\nabla\tilde{\ell}(\mathbf{M}_c^{(k)})\|_F^2 \ \leq \ \tilde{\ell}(\mathbf{M}_c^{(k)}) - \frac{\eta}{2}\|\nabla\tilde{\ell}(\mathbf{M}_c^{(k)})\|_F^2.$$

Using convexity of $\tilde{\ell}$, i.e. $\tilde{\ell}(\mathbf{M}_c^*) \geq \tilde{\ell}(\mathbf{M}_c^{(k)}) + \langle\nabla\tilde{\ell}(\mathbf{M}_c^{(k)}), \mathbf{M}_c^* - \mathbf{M}_c^{(k)}\rangle_F$, we have:

$$\tilde{\ell}(\mathbf{M}_c^{(k+1)}) \leq \tilde{\ell}(\mathbf{M}_c^{(k)}) + \langle\nabla\tilde{\ell}(\mathbf{M}_c^{(k)}), \mathbf{M}_c^{(k)} - \mathbf{M}_c^*\rangle_F - \frac{\eta}{2}\|\nabla\tilde{\ell}(\mathbf{M}_c^{(k)})\|_F^2$$

$$= \tilde{\ell}(\mathbf{M}_c^*) + \frac{1}{2\eta}\|\mathbf{M}_c^{(k)} - \mathbf{M}_c^*\|_F^2 - \frac{1}{2\eta}\|\mathbf{M}_c^{(k)} - \mathbf{M}_c^*\|_F^2 + \langle\nabla\tilde{\ell}(\mathbf{M}_c^{(k)}), \mathbf{M}_c^{(k)} - \mathbf{M}_c^*\rangle_F - \frac{\eta}{2}\|\nabla\tilde{\ell}(\mathbf{M}_c^{(k)})\|_F^2$$

$$= \tilde{\ell}(\mathbf{M}_c^*) + \frac{1}{2\eta}\|\mathbf{M}_c^{(k)} - \mathbf{M}_c^*\|_F^2 - \frac{1}{2\eta}\|\mathbf{M}_c^{(k)} - \mathbf{M}_c^* - \eta\nabla\tilde{\ell}(\mathbf{M}_c^{(k)})\|_F^2$$

$$= \tilde{\ell}(\mathbf{M}_c^*) + \frac{1}{2\eta}\left(\|\mathbf{M}_c^{(k)} - \mathbf{M}_c^*\|_F^2 - \|\mathbf{M}_c^{(k+1)} - \mathbf{M}_c^*\|_F^2\right). \tag{$*$}$$

Summing $(*)$ over $k = 0, \ldots, K - 1$:

$$\sum_{k=0}^{K-1}\left[\tilde{\ell}(\mathbf{M}_c^{(k+1)}) - \tilde{\ell}(\mathbf{M}_c^*)\right] \leq \frac{1}{2\eta}\left(\|\mathbf{M}_c^{(0)} - \mathbf{M}_c^*\|_F^2 - \|\mathbf{M}_c^{(K)} - \mathbf{M}_c^*\|_F^2\right) \leq \frac{\varepsilon_0^2}{2\eta}. \tag{16}$$

Since $(*)$ implies $\tilde{\ell}(\mathbf{M}_c^{(k+1)}) \leq \tilde{\ell}(\mathbf{M}_c^{(k)})$, we have $K\tilde{\ell}(\mathbf{M}_c^{(K)}) \leq \sum_{k=0}^{K-1}\tilde{\ell}(\mathbf{M}_c^{(k+1)})$, thus:

$$K\tilde{\ell}(\mathbf{M}_c^{(K)}) - K\tilde{\ell}(\mathbf{M}_c^*) \leq \sum_{k=0}^{K-1}\left[\tilde{\ell}(\mathbf{M}_c^{(k+1)}) - \tilde{\ell}(\mathbf{M}_c^*)\right] \leq \frac{\varepsilon_0^2}{2\eta}. \tag{17}$$

Therefore, $\tilde{\ell}\left(\mathbf{M}_c^{(K)}\right) - \tilde{\ell}(\mathbf{M}_c^*) \leq \frac{\varepsilon_0^2}{2\eta K}$, which gives the stated $O(1/K)$ rate. $\qquad\square$

Theorem A.1 establishes that the memory optimization converges at an $O(1/K)$ rate, confirming the stability of the gradient-based update at each streaming step.

*Table 7.* **Hand-crafted vs. GPT-4o attributes** (ViT-B/16). Accuracy (%) $\pm$ std over 20 random stream orderings.

| Method | 10% DTD | 100% DTD | 10% FGVC | 100% FGVC | 10% CD Avg | 100% CD Avg | 10% OOD Avg | 100% OOD Avg |
|---|---|---|---|---|---|---|---|---|
| CLIP zero-shot | 42.97±0.00 | 45.04±0.00 | 22.15±0.00 | 23.22±0.00 | 61.62±0.00 | 64.59±0.00 | 56.69±0.00 | 59.42±0.00 |
| TDA | 41.83±1.41 | 47.53±1.24 | 21.47±0.96 | 23.87±0.86 | 60.53±1.03 | 67.41±0.98 | 55.92±1.14 | 63.72±1.07 |
| ADAPT | 42.58±0.74 | 55.08±0.63 | 21.83±0.54 | 28.91±0.45 | 61.41±0.56 | 70.31±0.52 | 56.17±0.31 | 64.92±0.54 |
| Ours (hand-crafted) | 44.52±0.47 | 56.94±0.35 | 23.43±0.51 | 29.67±0.38 | 62.64±0.27 | 71.09±0.23 | 57.71±0.24 | 65.74±0.21 |
| Ours (GPT-4o) | **45.17±0.37** | **57.41±0.29** | **23.84±0.36** | **29.92±0.28** | **63.79±0.21** | **71.53±0.19** | **58.87±0.19** | **66.09±0.17** |

*Table 8.* **Hand-crafted vs. GPT-4o attributes** (RN50). Accuracy (%) $\pm$ std over 20 random stream orderings.

| Method | 10% DTD | 100% DTD | 10% FGVC | 100% FGVC | 10% CD Avg | 100% CD Avg | 10% OOD Avg | 100% OOD Avg |
|---|---|---|---|---|---|---|---|---|
| CLIP zero-shot | 38.51±0.00 | 40.37±0.00 | 15.37±0.00 | 16.11±0.00 | 54.03±0.00 | 56.63±0.00 | 41.11±0.00 | 43.09±0.00 |
| TDA | 37.24±1.45 | 43.61±1.31 | 14.89±0.92 | 17.58±0.81 | 53.17±1.07 | 60.87±1.02 | 40.38±1.12 | 46.47±1.08 |
| ADAPT | 38.19±0.67 | 51.76±0.58 | 15.21±0.61 | 17.97±0.52 | 53.86±0.37 | 62.52±0.34 | 40.94±0.38 | 47.68±0.35 |
| Ours (hand-crafted) | 40.08±0.52 | 51.86±0.38 | 16.31±0.54 | 19.64±0.42 | 55.47±0.31 | 63.28±0.27 | 42.53±0.27 | 48.47±0.24 |
| Ours (GPT-4o) | **40.69±0.38** | **52.03±0.31** | **16.74±0.44** | **19.91±0.33** | **56.08±0.22** | **63.68±0.19** | **43.14±0.23** | **48.82±0.21** |

## B. More Experiment Results

### B.1. Ablation on Attribute Construction: Hand-Crafted vs. GPT-4o

We replace GPT-4o attributes with standard hand-crafted templates (e.g., "a photo of a {class}") while keeping the full memory framework intact. Tables 7 and 8 report results under both backbones, evaluated at 10% and 100% of the test stream (averaged over 20 random orderings). Across all settings, *hand-crafted* consistently outperforms TDA and ADAPT even on texture-heavy (DTD) and fine-grained (FGVC) datasets where attribute decomposition is less straightforward. This confirms that the **associative memory mechanism is the primary driver of performance gains**, while GPT-4o attributes provide a complementary boost of ∼0.3–0.8% by capturing richer semantic structure.

*Table 9.* Top-1 accuracy (%) comparison on **Corruption Robustness** (ImageNet-C) under the Online protocol.

| Method | Blur | | | | Weather | | | | Digital | | | | Noise | | | Avg. |
|---|---|---|---|---|---|---|---|---|---|---|---|---|---|---|---|---|
| | Defo. | Glas. | Moti. | Zoom | Snow | Fros. | Fog | Brig. | Cont. | Elas. | Pix. | JPEG | Gauss. | Shot | Impu. | |
| CLIP-ViT-B/16 | 24.25 | 15.71 | 24.46 | 22.60 | 33.08 | 31.06 | 37.61 | 55.62 | 17.11 | 13.43 | 33.04 | 33.70 | 13.25 | 14.16 | 13.48 | 25.50 |
| TPT | _27.56_ | 15.48 | 26.16 | _26.94_ | _36.74_ | 34.28 | 39.38 | 60.22 | 16.96 | 15.64 | **40.74** | _37.90_ | 10.64 | 11.94 | 10.92 | 27.43 |
| TDA | 26.53 | 17.91 | _27.35_ | 25.90 | 36.50 | _34.84_ | 40.53 | 58.57 | _20.16_ | _16.62_ | 35.65 | 36.69 | 15.42 | 16.46 | _16.03_ | 28.34 |
| ADAPT | 26.30 | _18.01_ | 27.31 | 25.54 | 36.19 | 34.67 | _40.96_ | _60.29_ | 19.95 | 16.09 | 37.44 | 37.22 | _15.76_ | _16.84_ | 15.90 | _28.56_ |
| **Ours** | **28.15** | **18.55** | **28.42** | **27.88** | **37.50** | **36.12** | **41.50** | **61.15** | **21.05** | **17.20** | _39.50_ | **38.55** | **16.10** | **17.25** | **16.55** | **29.70** |
| CLIP-RN50 | 9.54 | 3.40 | 7.46 | 12.62 | 12.29 | 15.72 | 22.08 | 41.69 | 6.24 | 4.67 | 11.01 | 14.24 | 2.43 | 3.07 | 2.52 | 11.27 |
| TPT | 8.02 | 2.74 | 5.34 | 10.97 | 10.59 | 12.92 | 16.17 | 35.67 | 4.45 | 3.73 | 11.56 | **16.68** | 1.43 | 1.94 | 1.42 | 9.58 |
| TDA | 9.84 | 4.40 | 7.38 | 13.74 | _13.74_ | 17.16 | 23.76 | 44.16 | 7.00 | 5.79 | 11.24 | 15.26 | _2.54_ | 3.26 | 2.72 | 12.13 |
| ADAPT | _10.54_ | _4.44_ | **8.57** | _14.34_ | **13.85** | _17.84_ | **24.56** | _45.67_ | _7.76_ | _5.85_ | _11.96_ | _15.86_ | **2.91** | **3.77** | **2.92** | _12.72_ |
| **Ours** | **10.66** | **4.56** | _8.48_ | **15.26** | 12.67 | **19.67** | _24.53_ | **46.99** | **7.78** | **5.88** | **12.01** | 15.85 | 1.91 | _3.76_ | _2.88_ | **12.86** |

### B.2. Corruption Robustness

Table 9 reports the evaluation on ImageNet-C (Hendrycks & Dietterich, 2019), which contains 15 corruption types covering noise, blur, weather, and digital artifacts. Our A$^2$Memory consistently outperforms all baselines across both backbones. Specifically, with ViT-B/16, it achieves an average accuracy of **29.70%**, surpassing the SOTA method ADAPT (28.56%) and Zero-shot CLIP (25.50%) by substantial margins. Our method exhibits strong resilience in severe corruptions like *Weather* and *Blur*, owing to the stability of attribute-centric representations and gradient-based memory updates. This superiority extends to the ResNet-50 backbone (**12.86%** avg.), verifying that our framework is effective across different architectures.

## C. More Implementation Details

### C.1. Prompting Pipeline for Attribute-Centric Representation Construction

We describe the prompt engineering framework used to construct the prior textual representation $\mathbf{T}$ in Eq. (8). As detailed in Section 3.2, our goal is to generate $N$ shared visual attributes across $C$ categories and subsequently instantiate them into fine-grained descriptions. We utilize GPT-4o (Achiam et al., 2023) with a two-stage pipeline: (1) *Class-Shared Visual Attribute Generation* and (2) *Class-Specific Attribute Instantiation*.

---

**Protocol: Generating $N$ Attribute-Centric Descriptions for $C$ Classes**

STAGE 1: CLASS-SHARED ATTRIBUTE GENERATION
**Objective:** Identify a set of $N$ candidate visual attributes $\mathcal{A} = \{a_1, ..., a_N\}$ shared across the entire dataset $\mathcal{C}$.
**System Prompt:** You are a computer vision expert analyzing visual characteristics.
**User Input:**
- A list of $C$ class names: $\mathcal{C} = [\text{class}_1, \text{class}_2, \ldots, \text{class}_C]$.
- Target count: $N$ attributes.
**Task Instruction:** Generate exactly $N$ shared **VISUAL** attributes that can be observed in images and are useful for distinguishing between these classes.
- **Invariance:** Focus on visual features invariant to lighting, background, or camera angle.
- **Observability:** Focus on observable characteristics like shape, color, texture, size, and structure.
- **Distinctiveness:** Identify distinctive visual patterns or components.
STAGE 2: LOCAL ATTRIBUTE INSTANTIATION
**Objective:** Generate a detailed textual description $t_{c,n}$ for each class $c \in \mathcal{C}$ regarding attribute $n \in \{1, ..., N\}$.
**System Prompt:** You are a computer vision expert writing detailed visual descriptions.
**User Input:** A JSON object containing the specific class $c$ and the attribute list $\mathcal{A}$:

```
{
  "class_name": "c",
  "attributes": ["a_1", "a_2", ..., "a_N"]
}
```

**Task Instruction:** For each attribute $a_n$, write a detailed sentence describing how it appears in images of class $c$.
- **Concrete Features:** Focus on concrete visual characteristics that would be visible in photos.
- **Specific Details:** Include specific details about appearance (colors, shapes, textures, structures).
- **Consistency:** Focus on features that remain consistent across different viewing conditions.
- **Discrimination:** Highlight observable patterns that help identify this class.
- **Style Constraint:** Write naturally varied descriptions - **do NOT** use rigid templates.
**Output:** A JSON dictionary mapping attributes to descriptions:

```
{
  "descriptions": {
    "a_1": "Description of a_1 for class c...",
    ...
    "a_N": "Description of a_N for class c..."
  }
}
```

---

To illustrate the pipeline, we provide a concrete example using the Oxford-Pets dataset (Parkhi et al., 2012). The model first identifies 10 shared visual attributes valid across cat and dog categories, and then instantiates specific descriptions for the *Abyssinian* class. Note that in Stage 2, the generated descriptions explicitly reference the class name to ensure semantic context.

---

**Example Output: Oxford-Pets Generation**

---

STAGE 1: SHARED ATTRIBUTE GENERATION
**User Input (Subset):**

```
Classes: ['Abyssinian', 'American Bulldog', ..., 'Yorkshire Terrier']
Target: 10 Attributes
```

**Model Output (Generated Shared Attributes $\mathcal{A}$):**
  1. **Head Shape** (Visual structure of the skull and face)
  2. **Ear Structure** (Shape, size, and set of the ears)
  3. **Muzzle Form** (Length and width of the snout/jaw)
  4. **Body Build** (Overall muscularity and skeletal structure)
  5. **Coat Texture** (Tactile and visual quality of the fur)
  6. **Tail Carriage** (Shape and positioning of the tail)
  7. **Leg Length** (Proportion of limbs to body)
  8. **Eye Shape** (Geometry of the eye opening)
  9. **Fur Pattern** (Distribution of color markings)
 10. **Paw Size** (Relative size and shape of feet)

STAGE 2: ATTRIBUTE INSTANTIATION
**User Input:**

```
{
  "class_name": "Abyssinian",
  "attributes": ["Coat Texture", "Ear Structure"]
}
```

**Model Output (Generated Descriptions):**

```
{
  "descriptions": {
    "Coat Texture": "The Abyssinian features a short, fine coat that is close
      -lying to the body, exhibiting a resilient texture with a distinct ticked
      pattern that gives it a lustrous sheen.",

    "Ear Structure": "The ears of the Abyssinian are large, alert, and
      moderately pointed at the tips, broadly cupped at the base and set wide
      apart."
  }
}
```

---

## C.2. Benchmark Details

In this section, we elaborate on the two benchmarks utilized in our experiments: the Out-of-Distribution (OOD) Benchmark and the Cross-Domain Benchmark.

The **OOD Benchmark** is employed to assess model robustness against natural distribution shifts. It comprises the standard ImageNet (Deng et al., 2009) validation set along with four OOD variants: ImageNet-V2 (Recht et al., 2019), ImageNet-A (Hendrycks et al., 2021b), ImageNet-R (Hendrycks et al., 2021a), and ImageNet-S (Wang et al., 2019), covering various corruptions, styles, and sketches. The **Cross-Domain Benchmark** evaluates generalization across 10 diverse datasets, ranging from fine-grained classification (e.g., OxfordPets (Parkhi et al., 2012), StanfordCars (Krause et al., 2013)) to specialized domains like texture (DTD) and satellite imagery (EuroSAT) (Helber et al., 2019).

The detailed statistics for all datasets, including class counts and test set sizes, are summarized in Table 10.

*Table 10.* Detailed statistics of the evaluated datasets.

| Dataset | Task | Classes | Training Size | Testing Size |
|---|---|---|---|---|
| ImageNet | Object recognition | 1000 | 1.28M | 50,000 |
| Caltech101 | Object recognition | 100 | 4,128 | 2,465 |
| OxfordPets | Fine-grained pets recognition | 37 | 2,944 | 3,669 |
| StanfordCars | Fine-grained car recognition | 196 | 6,509 | 8,041 |
| Flowers102 | Fine-grained flowers recognition | 102 | 4,093 | 2,463 |
| Food101 | Fine-grained food recognition | 101 | 50,500 | 30,300 |
| FGVCAircraft | Fine-grained aircraft recognition | 100 | 3,334 | 3,333 |
| SUN397 | Scene recognition | 397 | 15,880 | 19,850 |
| DTD | Texture recognition | 47 | 2,820 | 1,692 |
| EuroSAT | Satellite image recognition | 10 | 13,500 | 8,100 |
| UCF101 | Action recognition | 101 | 7,639 | 3,783 |
| ImageNet-V2 | Robustness of collocation | 1000 | N/A | 10,000 |
| ImageNet-Sketch | Robustness of sketch domain | 1000 | N/A | 50,889 |
| ImageNet-A | Robustness of adversarial attack | 200 | N/A | 7,500 |
| ImageNet-R | Robustness of multi-domains | 200 | N/A | 30,000 |

## C.3. Pseudo Code.

The pseudo code of $\mathbf{A^2Memory}$ is illustrated in Algorithm 1.

---

**Algorithm 1** $A^2$Memory for Test-time Adaptation

---

**Require:** Streaming test data $\mathcal{D}_{\text{test}} = \{\boldsymbol{x}_1, \boldsymbol{x}_2, \dots\}$, Pre-trained CLIP model $\{f_\theta, g_\psi\}$, LLM.
**Require:** Hyperparameters: Candidate size $N$, Selection size $L$, Balance factor $\lambda$, Learning rate $\eta$.
 1: **// Phase 1: Attribute-centric Representation Construction (Offline)**
 2: Generate $N$ shared class-agnostic attributes via LLM.
 3: Instantiate class-specific descriptions $\{t_{c,n}\}$ and encode to $\mathbf{T}^{\text{can}}$.
 4: Initialize sets: $\mathcal{S}_{\text{sel}} \leftarrow \emptyset, \mathcal{S}_{\text{can}} \leftarrow \{1, \dots, N\}$.
 5: **while** $|\mathcal{S}_{\text{sel}}| < L$ **do**
 6:     Compute $\text{Rep}(k)$ and $\text{Div}(k)$ for all $k \in \mathcal{S}_{\text{can}}$ via Eq. (5).
 7:     Select attribute $k^* = \arg\max_{k \in \mathcal{S}_{\text{can}}}[\lambda \cdot \text{Rep}(k) + (1 - \lambda) \cdot \text{Div}(k)]$.
 8:     Update sets: $\mathcal{S}_{\text{sel}} \leftarrow \mathcal{S}_{\text{sel}} \cup \{k^*\}, \mathcal{S}_{\text{can}} \leftarrow \mathcal{S}_{\text{can}} \setminus \{k^*\}$.
 9: **end while**
10: Construct Prior Textual Representation $\mathbf{T} \in \mathbb{R}^{C \times L \times D}$ via Eq. (8).
11: Initialize class-wise associative memories $\{\mathbf{M}_c^0 = \mathbf{0}\}_{c=1}^C$ and retention coefficient $\alpha_0 = 0$.
12: **// Phase 2: Online Test-time Adaptation**
13: **for** time step $t = 1, 2, \dots$ **do**
14:     Receive test image $\mathbf{x}_t$.
15:     Extract visual feature $\boldsymbol{f}^t = \text{norm}(f_\theta(\boldsymbol{x}_t))$.
16:     *// Surrogate Visual Representation Construction*
17:     **for** each class $c$ and attribute $l \in \{1, \dots, L\}$ **do**
18:         Compute $\mathbf{I}_{c,l}^t = \text{norm}(\boldsymbol{f}^t + \text{norm}(\boldsymbol{f}^t \odot \sigma(\boldsymbol{f}^t \odot \mathbf{T}_{c,l})))$.
19:     **end for**
20:     *// Class-wise Associative Memory Optimization*
21:     **for** each class $c$ **do**
22:         Compute Loss: $\ell = \frac{1}{L}\sum_l \|\mathbf{M}_c^{t-1} \cdot \phi(\mathbf{I}_{c,l}^t) - \mathbf{T}_{c,l}\|_2^2$.
23:         Compute Gradient $\nabla_{\mathbf{M}_c}$ via Eq. (11).
24:         Update Memory: $\mathbf{M}_c^t = \alpha_{t-1}\mathbf{M}_c^{t-1} + \eta\nabla_{\mathbf{M}_c}$ via Eq. (12).
25:     **end for**
26:     *// Inference via Memory Retrieval & Fusion*
27:     Compute CLIP logits: $p_c^{\text{clip}} = \langle \boldsymbol{f}^t, \frac{1}{L}\sum_l \mathbf{T}_{c,l}\rangle$.
28:     Query Memory: $\boldsymbol{o}_c^t = \phi(\boldsymbol{f}^t)^\top \mathbf{M}_c^t$.
29:     Compute Memory logits: $p_c^{mem} = \langle \boldsymbol{o}_c^t, \boldsymbol{f}^t\rangle$ via Eq. (14).
30:     Fuse predictions: $\boldsymbol{P}_t^{final} = \lambda^{clip}\boldsymbol{P}_t^{clip} + \lambda^{mem}\boldsymbol{P}_t^{mem}$ via Eq. (15).
31:     Output prediction $\hat{y}_t = \arg\max \boldsymbol{P}_t^{final}$.
32:     Update retention coefficient $\alpha_t$ based on entropy $\tilde{H}(\boldsymbol{P}_t^{final})$ via Eq. (13).
33: **end for**

---

