# OpenReview forum: "Learning to Memorize with Attributive and Associative Memory for Online Test-Time Adaptation of Vision-Language Models"
_ICML.cc/2026/Conference — ICML 2026 regular_

### Official Review · Reviewer_9srp · 2026-03-05

**Soundness:** 2
**Presentation:** 2
**Significance:** 3
**Originality:** 3
**Overall Recommendation:** 5
**Confidence:** 4

**Summary:**

- authors propose 3 contributions
	- in tta, assigning one class to each item leads to instability, so there needs to be a mechanism for multi-class assignment
	- rule based memory eviction is not always the best choice since it destroys representation quality
  - A2 memory uses soft-assignment to retrieve class prototypes

**Compliance With Llm Reviewing Policy:**

Affirmed.

**Final Justification:**

I like this,

please release code,

novelty:

it's novel in TDA.



prior work:

acknowledge delta rule,

acknowledge ttt-memory compression ,

add future work

🖖

----

**Key Questions For Authors:**

please see weakness.

**Limitations:**

- Please discuss limitations in conclusion.
i could not find them

**Strengths And Weaknesses:**

Strengths

  - indeed the equation 3 is super interesting, it reminds me of ttt layers
	- where matrix M is some network, query is projected with kernel phi, and learning compresses k-v cache into weights of the net.
	- given this step of gradient descent, do authors have any suggestions how we could make it more efficient than heuristics?

- Indeed, equation 5 looks interesting.
    - The idea of  prompting llms to get multiple attributes, and use them for controlling representativeness and diversity indeed seems super interesting.
  - Ablations are well informed, in particular

Limitations.

I believe this paper offers a novel contribution to ICML, since the idea of controlling representativeness, and compressing memory info via gradient updates seems new in TTA (to the best of my knowledge TTT-MLP does propose a similar idea). The question whether TTA/TTT bear resemblance, is a `sensitive' one, therefore i will refrain  from that.

Among other ideas proposed here, the ability to share information among different classes, is indeed interesting, since objects in nature can form with a mixture of attributes, and methods like clip treat them as independent classes, which does not explain zero-shot-compositionally. I believe that ideas from this paper may be of some significance to general representational learning. Although, it works for classification, it would be interesting to see how it holds upto segmentation.


- What happens if llm cannot give multiple attributes?
- No of class slots scale linearly with no of classes. What happens there are 1k classes. Or more?
- Table 5, should have tent method
- Also I am not sure of tda time, there paper claims 16 min on imagenet with resnet50 (https://github.com/kdiAAA/TDA).
    - But it seems the paper reports ~1hr, which is same as what DPE reported.
    - So I think there is some confusion somewhere, could the authors please tell what there own experiments have revealed,
	- Please discuss limitations in conclusion

In summary, this paper did teach me something new, and i am grateful to the authors for their submission.

----
increased rating post-rebuttal.


I thank the authors for their rebuttal.

—

I agree with Reviewer u92A that size of vocabulary |V|, and associative matrix (D by D), scales with classes, presents a memory bottleneck beyond TDA.

To me, a deeper reason seems that the attributes still need to be brought into existence (line 131, para is second column) . An ability to directly operate on weights (of text encoder), without estimating neural activations would indeed be interesting[1].

Indeed, One may eventually may compress neural activities (keys/values)  in weights of a model (like this paper does).

However, it is unfair, to penalize the paper on it, since most sota (including ttt layers) appears to lie here.

—

Indeed, as reviewer aPYZ, points out, results in table 1, are low on several datasets,

I agree this is very important.  Authors should discuss this, and duly acknowledge this limitation.

I still appreciate that average score is better than other baselines .

——

Personally, I believe that the application of delta rule to adaptation  is interesting. A similar delta-rule was also applied earlier by Songlin to GatedDeltaNet[2], and I think this may carry implications further.

My own  concerns thus appear addressed, I  gladly revise my rating to accept.


—
[1] https://arxiv.org/abs/1610.06258
[2] https://arxiv.org/abs/2412.06464

---

> ### Author Rebuttal · Authors · 2026-03-30
>
> **Thank you very much for your encouraging feedback and for recognizing our interesting formulations, LLM attribute prompting strategy, and well-informed ablations. Below are our responses to your specific concerns.**
> > **W1:** suggestions how we could make it more efficient?
>
> **Response:** Parameterizing the memory matrix in a LoRA-style low-rank form is a promising direction to reduce per-step update cost. Exploring efficient online optimization strategies is a rich avenue for future work.
>
> > **W2:**What happens if llm cannot give multiple attributes?
>
> **Response:** Attribute generation is an offline one-time step. If unavailable, it falls back to standard hand-crafted templates. **We add experiments comparing GPT-4o attributes vs. hand-crafted templates to demonstrate robustness.** Even with hand-crafted templates, our method outperforms ADAPT, confirming that the **memory mechanism itself is the primary driver** of performance gains. GPT-4o attributes and the memory mechanism are **complementary and both valuable**.
>
> | Method | ImageNet |
> |--------|----------|
> | CLIP zero-shot | 68.34 ± 0.00 |
> | TDA | 69.51 ± 1.12 |
> | ADAPT | 70.91 ± 0.32 |
> | **hand-crafted** | **71.43 ± 0.24** |
> | **Ours (GPT-4o)** | **72.21 ± 0.18** |
>
> > **W3:** What happens there are 1k classes. Or more?
>
> **Response:** We find that memory usage is a bottleneck (e.g., ~262M parameters for ImageNet vs. TDA's ~5.1M). To address this, we propose a shared-memory: all classes share one global matrix $\mathbf{M}\_{\text{shared}} \in \mathbb{R}^{D \times D}$, while each class maintains a lightweight bias $\mathbf{b}\_c \in \mathbb{R}^D$. Let ${\phi}\_{c,l} = {\phi}({\mathbf{I}\_{c,l}^t})$, The objective becomes: $\ell_c = \frac{1}{L}\sum_{l} ||\mathbf{M}_{\text{shared}}\phi\_{c,l} + \mathbf{b}_c - \mathbf{T}_l||_2^2$.
>
> The resulting online update rules are: $\mathbf{b}\_c^t = \alpha\_{t-1}\mathbf{b}\_c^{t-1} - \eta\_b \cdot \frac{2}{L}\sum\_l \mathbf{r}\_{c,l}$ and $\mathbf{M}\_{\text{shared}}^t = \alpha\_{t-1}\mathbf{M}\_{\text{shared}}^{t-1} - \eta\_M \cdot \frac{2}{CL}\sum\_c\sum_l \mathbf{r}\_{c,l}\phi\_{c,l}^\top$, where $\mathbf{r}\_{c,l} = \mathbf{M}\_{\text{shared}}\phi\_{c,l} + \mathbf{b}\_c - \mathbf{T}\_l$. The shared matrix captures universal visual-semantic mappings while per-class biases absorb class-specific offsets. As the number of classes scales further, the memory footprint of this variant grows only linearly with the lightweight bias vectors ($O(C \cdot D)$) rather than the full matrices ($O(C \cdot D^2)$)
>
> **We add experiments on the shared-memory variant.**  Ours achieves the best accuracy overall. The variant achieves comparable performance while reducing parameters to ~0.77M, offering a much more memory-efficient alternative.
> | Method | ImageNet | OOD Avg | CD Avg |
> |--------|----------|---------|--------|
> | ADAPT | 70.72 ± 0.32 | 64.92 ± 0.28 | 70.31 ± 0.52 |
> | **Ours** | **72.28 ± 0.18** | **66.09 ± 0.17** | **71.53 ± 0.19** |
> | **shared M + bias** | **72.19 ± 0.22**| **65.78 ± 0.13** | **71.14 ± 0.17** |
>
> > **W4:** Table 5 should have tent method
>
> **Response:** **We add experiments comparing with Tent under the fully online, batch-size-1 setting.** Tent relies on updating batch normalization statistics, which is fundamentally ill-suited for this setting. Tent suffers a catastrophic performance drop (22.42%) due to unstable BN statistics at batch size 1, while also taking significantly longer, making it impractical for deployment.
>
>
> | Method | ImageNet Accuracy (%) | Inference Time (min) |
> |--------|-----------------------|----------------------|
> | CLIP zero-shot | 59.81 | 9.25 |
> | Tent | 22.42 | 125.00 |
> | **Ours** | **64.05** | **61.24** |
>
>
> > **W5:** There is a discrepancy in the reported inference time for TDA ; clarification on experimental findings is needed.
>
> **Response:**  The gap stems from `AugMix`[1] data augmentation in TDA's codebase, which generates 64 views and requires 64 forward passes per image by default. TDA's reported ~16 mins likely reflects 1 view.
> **We reported times (consistent with DPE and ADAPT) using 64 views for fair comparison**. Both methods still achieve comparable accuracy with 1 view, demonstrating that they do not heavily rely on AugMix. We recommend setting views to 1 for practical efficiency.
>
>
> | Method | AugMix Views | Time (min) | Accuracy (%) |
> |--------|--------------|----------------------|--------------|
> | TDA | 1 | 13.56 | 69.42 |
> | TDA | 64 | 52.50 | 69.51 |
> | **Ours** | 1 | 16.72 | 72.19 |
> | **Ours** | 64 | 69.24 | 72.21 |
>
> > **W6:** Please discuss limitations in conclusion.
>
> **Response:** We will add a Limitations section discussing: (1) **Efficiency**: Inference speed can be further improved; (2) **Task Scope**: Extending to dense prediction tasks remains open; (3) **Hyperparameters**: Exploring fully adaptive mechanisms is a promising future direction.
>
> Reference
>
>   1.“Augmix: A simple data processing method to improve robustness and uncertainty. In ICLR 2020.

---

> > ### Author Rebuttal · Reviewer_9srp · 2026-04-01
> >
> > Thanks for the detailed rebuttal, my questions are resolved. I have increased my rating.

---

> > > ### Author Response · Authors · 2026-04-02
> > >
> > > **Thank you for raising your rating from 4 to 5, as well as the detailed post-rebuttal discussion and engaging with the concerns raised by Reviewers u92A and aPYZ.**
> > >
> > > > **(Engaging with Reviewer u92A) An ability to directly operate on weights (of text encoder), without estimating neural activations would indeed be interesting. However, it is unfair, to penalize the paper on it, since most sota (including ttt layers) appears to lie here.**
> > >
> > > **Response:** Thank you for suggesting this direction, as well as helping us to clarify most sota appears to have this problem. We acknowledge that our method still relies on neural activations. Directly operating on encoder weights is an inspiring direction, which could eliminate the feature-extraction bottleneck and enable richer representational adaptation. We intend to pursue this direction in our future research.
> > >
> > > > **(Engaging with Reviewer aPYZ) Results in table 1, are low on several datasets. Authors should discuss this, and duly acknowledge this limitation. I still appreciate that average score is better than other baselines.**
> > >
> > > **Response:** We acknowledge no single method dominates every datasets, which is common in TTA [1, 2], and will discuss this in the Limitations section. Please kindly note that our evaluation covers all 15 benchmarks.
> > >
> > > To fully address this concern, we also **add experiments to evaluate the #Top-k metric, measured by the fraction of datasets (out of 15) where a method achieves top-k accuracy.**
> > >
> > >
> > > | Method | #Top-1 | #Top-2 | #Top-3 |
> > > |--------|--------|--------|--------|
> > > | **A²Memory — ViT-B/16** | **9**/15 | **13**/15 | **15**/15 |
> > > | **A²Memory — RN50** | **8**/15 | **13**/15 | **13**/15 |
> > > | ADAPT — ViT-B/16 | 1/15 | 8/15 | 12/15 |
> > > | ADAPT — RN50 | 3/15 | 6/15 | 11/15 |
> > > ||
> > >
> > > From above, **A²Memory consistently outranks ADAPT across all tiers, demonstrating the strongest overall generalization.**
> > >
> > > > **Personally, I believe that the application of delta rule to adaptation is interesting. A similar delta-rule was also applied earlier by Songlin to GatedDeltaNet[2], and I think this may carry implications further.**
> > >
> > > **Response:** We are delighted the connection resonated. Our motivation stems from its compact compression without eviction, gradient-driven selectivity, and data-dependent plasticity-stability balance. We look forward to exploring further in this direction [3, 4].
> > >
> > > ***
> > >
> > > We really appreciate your recognition of our work and your kind words, and we are happy to hear that your concerns have been adequately addressed. Again, thank you for your valuable suggestions which have undoubtedly contributed to improving the quality of our paper.
> > >
> > > ***
> > >
> > > **References**
> > >
> > > [1] Backpropagation-Free Test-Time Adaptation via Probabilistic Gaussian Alignment.
> > >
> > > [2] Bayesian Test-Time Adaptation for Vision-Language Models.
> > >
> > > [3] Lattice: Learning to Efficiently Compress the Memory.
> > >
> > > [4] It’s All Connected: A Journey Through Test-Time Memorization, Attentional Bias, Retention, and Online Optimization.

---

### Official Review · Reviewer_ZW59 · 2026-03-06

**Soundness:** 3
**Presentation:** 3
**Significance:** 3
**Originality:** 3
**Overall Recommendation:** 5
**Confidence:** 3

**Summary:**

This paper proposes A2Memory, a memory-based framework for Test-Time Adaptation. The method addresses pseudo-label noise and biased memory eviction by introducing attribute-centric memory construction with soft assignment and a class-wise associative memory that compresses streaming samples within fixed capacity. Experiments on 15 benchmarks demonstrate consistent improvements over existing methods.

**Compliance With Llm Reviewing Policy:**

Affirmed.

**Key Questions For Authors:**

See weakness for details.

**Limitations:**

See weakness for details.

**Strengths And Weaknesses:**

Strengths:

1. The paper presents a clear motivation and is well written. It studies a meaningful and interesting problem in Test-Time Adaptation, making the overall contribution easy to understand.

2. The experimental results are convincing and demonstrate consistent improvements across multiple benchmarks.

3. The design of associative memory in A²Memory is an interesting idea that may inspire future research on memory-based TTA methods.

Weaknesses:

1. The convergence behavior of the gradient-based optimization used in the associative memory module under a non-convex optimization setting is not discussed and would benefit from further analysis.

2. The t-SNE visualization in Figure 1 only shows the sample distribution selected by entropy-based strategies but does not compare the memory feature distribution produced by A²Memory, making it difficult to directly observe improvements in feature-space coverage.

3. Figure 2 does not clearly highlight the core innovations of the proposed method, making it harder for readers to quickly grasp the main technical contributions.

4. The overall technical pipeline appears relatively complex, and the reproducibility of the method may rely on the availability of the implementation code.

---

> ### Author Rebuttal · Authors · 2026-03-30
>
> **Thank you very much for your positive assessment and for recognizing our clear motivation, convincing results, and the interesting associative memory design. Below are our responses to your specific concerns.**
>
> > **W1:** The convergence behavior of the gradient-based optimization  lacks discussion.
>
> **Response:** We provide a formal convergence guarantee for our optimization objective:
>
> **Setup.** Let $\phi_l := \phi(\mathbf{I}_{c,l}^t)$, $||\phi_l||_2=1$, $\mathbf{t}\_l := \mathbf{T}\_{c,l}$. Eq. 9 optimizes $\tilde\ell(\mathbf{M}\_c) = \ell(\mathbf{M}\_c) + \frac{1-\alpha\_{t-1}}{2\eta}||\mathbf{M}_c||_F^2$. Let $\mathbf{M}\_c^* = \arg\min\tilde\ell$, $\varepsilon_0 = ||\mathbf{M}_c^{(0)} - \mathbf{M}_c^*||_F$, $\alpha\_{t-1}\in(0,1)$, $\eta>0$.
> 1. **Smoothness.** $\nabla\tilde\ell$ is $\tilde{L}$-Lipschitz, $\tilde{L} = 2 + \frac{1-\alpha_{t-1}}{\eta}$. For any $\mathbf{A},\mathbf{B}$:
> $||\nabla\tilde\ell(\mathbf{A})-\nabla\tilde\ell(\mathbf{B})||_F \leq \left(2+\frac{1-\alpha}{\eta}\right)||\mathbf{A}-\mathbf{B}||_F$.
> 2. **Initialization.** $||\mathbf{M}_c^{(0)}-\mathbf{M}_c^*||_F = \varepsilon_0 < \infty$.
>
> **Theorem.** Using conditions above, for all $K > 0$: $\tilde\ell\left(\mathbf{M}\_c^{(K)}\right) - \tilde\ell(\mathbf{M}\_c^\*) \leq \frac{\varepsilon\_0^2}{2\eta K}$.
>
> *Proof.*
>
>  **Convexity.** At update time, $\phi_l$ and $\mathbf{t}_l$ are fixed. Thus, $||\mathbf{M}_c\phi_l - \mathbf{t}_l||_2^2$ is the composition of an affine map with a squared norm, making $\tilde\ell$ a **convex quadratic** in $\mathbf{M}_c$.
>
> Since $\tilde\ell$ is convex and $\tilde{L}$-smooth, applying the descent lemma to $\mathbf{M}\_c^{(k+1)} = \mathbf{M}\_c^{(k)} - \eta\nabla\tilde\ell(\mathbf{M}\_c^{(k)})$ and setting $\eta \leq 1/\tilde{L}$:
>
> $$\tilde\ell(\mathbf{M}\_c^{(k+1)}) \leq \tilde\ell(\mathbf{M}\_c^{(k)}) + \langle\nabla\tilde\ell(\mathbf{M}\_c^{(k)}),\,\mathbf{M}\_c^{(k+1)}-\mathbf{M}\_c^{(k)}\rangle\_F + \frac{\tilde{L}}{2}||\mathbf{M}\_c^{(k+1)}-\mathbf{M}\_c^{(k)}||\_F^2$$
>
> $$= \tilde\ell(\mathbf{M}\_c^{(k)}) - \eta||\nabla\tilde\ell(\mathbf{M}\_c^{(k)})||\_F^2 + \frac{\tilde{L}\eta^2}{2}||\nabla\tilde\ell(\mathbf{M}\_c^{(k)})||\_F^2$$
>
> $$= \tilde\ell(\mathbf{M}\_c^{(k)}) - \eta\left(1 - \frac{\tilde{L}\eta}{2}\right)||\nabla\tilde\ell(\mathbf{M}\_c^{(k)})||\_F^2 \leq \tilde\ell(\mathbf{M}\_c^{(k)}) - \frac{\eta}{2}||\nabla\tilde\ell(\mathbf{M}\_c^{(k)})||\_F^2$$
>
> $$\leq \tilde\ell(\mathbf{M}\_c^\*) + \langle\nabla\tilde\ell(\mathbf{M}\_c^{(k)}),\mathbf{M}\_c^{(k)}-\mathbf{M}\_c^\*\rangle\_F - \frac{\eta}{2}||\nabla\tilde\ell(\mathbf{M}\_c^{(k)})||\_F^2$$
>
> $$= \tilde\ell(\mathbf{M}\_c^\*) + \frac{1}{2\eta}||\mathbf{M}\_c^{(k)}-\mathbf{M}\_c^\*||\_F^2 - \frac{1}{2\eta}||\mathbf{M}\_c^{(k)}-\mathbf{M}\_c^\*||\_F^2 + \langle\nabla\tilde\ell(\mathbf{M}\_c^{(k)}),\,\mathbf{M}\_c^{(k)}-\mathbf{M}\_c^\*\rangle\_F - \frac{\eta}{2}||\nabla\tilde\ell(\mathbf{M}\_c^{(k)})||\_F^2$$
>
> $$= \tilde\ell(\mathbf{M}\_c^\*) + \frac{1}{2\eta}||\mathbf{M}\_c^{(k)}-\mathbf{M}\_c^\*||\_F^2 - \frac{1}{2\eta}\bigl||\mathbf{M}\_c^{(k)}-\mathbf{M}\_c^\* - \eta\nabla\tilde\ell(\mathbf{M}\_c^{(k)})\bigr||\_F^2$$
>
> $$= \tilde\ell(\mathbf{M}\_c^\*) + \frac{1}{2\eta}\Bigl(||\mathbf{M}\_c^{(k)}-\mathbf{M}\_c^\*||\_F^2 - ||\mathbf{M}\_c^{(k+1)}-\mathbf{M}\_c^\*||\_F^2\Bigr) \quad (*)$$
>
> Summing $(*)$ over $k = 0, \ldots, K-1$:
>
> $$\sum\_{k=0}^{K-1}[\tilde\ell(\mathbf{M}\_c^{(k+1)})-\tilde\ell(\mathbf{M}\_c^\*)] \leq \frac{1}{2\eta}\Bigl(||\mathbf{M}\_c^{(0)}-\mathbf{M}\_c^\*||\_F^2 - ||\mathbf{M}\_c^{(K)}-\mathbf{M}\_c^\*||\_F^2\Bigr) \leq \frac{\varepsilon\_0^2}{2\eta}$$
>
> Since $(*)$ implies $\tilde\ell(\mathbf{M}\_c^{(k+1)}) \leq \tilde\ell(\mathbf{M}\_c^{(k)})$, we have $K\tilde\ell(\mathbf{M}\_c^{(K)}) \leq \sum\_{k=0}^{K-1}\tilde\ell(\mathbf{M}\_c^{(k+1)})$, thus:
>
> $$K\tilde\ell(\mathbf{M}\_c^{(K)}) - K\tilde\ell(\mathbf{M}\_c^\*) \leq \sum\_{k=0}^{K-1}\bigl[\tilde\ell(\mathbf{M}\_c^{(k+1)})-\tilde\ell(\mathbf{M}\_c^\*)\bigr] \leq \frac{\varepsilon\_0^2}{2\eta}$$
>
> Therefore, $\tilde\ell\left(\mathbf{M}\_c^{(K)}\right) - \tilde\ell(\mathbf{M}\_c^\*) \leq \frac{\varepsilon\_0^2}{2\eta K}$. $\square$
>
> > **W2:** Figure 1's t-SNE lacks a comparison with A²Memory's feature distribution.
>
> **Response:** **We add a new t-SNE visualization** (https://anonymous.4open.science/r/tta/2.png) showing TDA's features collapse into narrow clusters, while A²Memory's prototypes spread uniformly across each class, confirming broader coverage.
>
> > **W3:** Figure 2 fails to clearly highlight the core innovations.
>
> **Response:** **We have revised Figure 2** to better highlight the two core modules (https://anonymous.4open.science/r/tta/1.png), and will update it in the revision.
>
> > **W4:** The technical pipeline is complex, and reproducibility may depend on releasing the  code.
>
> **Response:** To support reproducibility, LLM prompts and detailed pseudocode are in the Appendix. We will release the full codebase upon acceptance.

---

### Official Review · Reviewer_u92A · 2026-03-11

**Soundness:** 2
**Presentation:** 2
**Significance:** 2
**Originality:** 3
**Overall Recommendation:** 4
**Confidence:** 4

**Summary:**

This paper addresses two limitations of memory-based test-time adaptation (TTA) for CLIP: (1) hard pseudo-label assignment contaminates class-specific memory slots, and (2) heuristic sample eviction under fixed capacity introduces selection bias. The proposed framework replaces explicit sample storage with two components: Attribute-centric Representation Construction, which uses LLM-generated visual attributes and text-guided gating to produce surrogate visual representations via soft assignment, and Class-wise Associative Memory, which parameterizes each class memory as a matrix updated by a delta-rule-style gradient step with entropy-adaptive retention. The proposed method is evaluated on 10 cross-dataset benchmarks and 5 OOD datasets using ViT-B/16 and RN50 backbones, reporting the best average accuracy across all settings.

**Compliance With Llm Reviewing Policy:**

Affirmed.

**Final Justification:**

I thank the authors for the constructive rebuttal. Most of my concerns have been substantially addressed, and I appreciate the additional experiments provided. Therefore, I will raise my rating.

**Key Questions For Authors:**

Please refer to Weaknesses above.

**Limitations:**

The paper should explicitly discuss practical limitations such as the memory cost, as well as the dependence on LLM, neither of which is currently acknowledged.

**Strengths And Weaknesses:**

* Strengths
	* To the best of my knowledge, this is the first work to apply delta-rule-based parametric associative memory (originally developed in the sequence modeling literature) to vision-language test-time adaptation. Reframing the memory from a discrete sample store to a continuously updated key-value matrix is a conceptually interesting direction that could inspire further work in this problem.
	* The two failure modes (memory contamination from hard assignment and selection bias from heuristic eviction) are clearly articulated and supported by targeted empirical analysis in Fig. 1. The observation that existing memory-based methods underperform zero-shot CLIP in early adaptation stages is a useful diagnostic that highlights a current limitation.
	* The experimental coverage is reasonable: 15 benchmarks, two CLIP backbones, fully only batch-size-1 evaluation, component ablations, and corruption robustness in the appendix.
* Weaknesses
	* Section 2.3 states the memory is optimized via gradient descent, yet Eq. (12) is gradient ascent if taken literally. This sign convention is repeated consistently in Algorithm 1 line 24, making it unclear whether this is an intentional design or a systematic typo. Furthermore, Algorithm 1 updates $\alpha_t$ using $P_t^{final}$ (line 26) before it is computed (line 31), and queries $M_{t-1}$ (line 29) after updating $M_t$ (line 24). The initialization of $M_0$ and $\alpha_0$ is also unspecified. These inconsistencies make the method unreproducible from the paper alone.
	* A substantial part of the method depends on a two-stage GPT-4o pipeline for attribute generation and class-specific descriptions. However, no ablation isolates this contribution. The component ablation (Table 3) removes $T$ entirely but does not replace it with a simpler alternative, so it conflates the benefit of richer text representation with the benefit of the associative memory formulation. This is particularly problematic because the early-stage performance advantage (Fig. 4, left) may largely stem from the enriched textual representations rather than the memory mechanism itself. Even without associative memory, the attribute-averaged text features may already improve substantially over single-template zero-shot CLIP.
	* The margins over the strongest baseline are mostly below one percentage point: 71.41 vs 70.76 on cross-dataset and 67.23 vs 66.53 on OOD for ViT-B/16, with similar margins for RN50. No variance across random seeds, stream-order sensitivity, or statistical significance tests are reported. Given the fully online, single-pass evaluation where stream ordering can affect accumulated memory states, these margins are insufficient to claim consistent superiority without repeated runs.
	* Each class maintains a $D\times D$ associative memory matrix. For ImageNet with $C=1000$ and $D=512$ (ViT-B/16), this amounts to approximately 262M parameters, which can exceed the memory footprint of sample-level methods like TDA. This is a significant limitation for a method that claims to be a practical, lightweight alternative. This point should be discussed.
	* The balancing factor $\lambda=0.3$ is fixed across all experiments, but Fig. 3(b) shows that the method is not robust to this choice: at $\lambda=0.1$ or $\lambda=0.5$, performance drops below that of competing baselines, meaning that a suboptimal $\lambda$ can make the method worse than the baselines. This turns $\lambda$ into a critical hyperparameter, yet the paper does not state whether it was tuned on a held-out validation set or the test benchmarks is not stated. If dataset-specific tuning is required, this conflicts with the online TTA assumption of no prior access to target distribution statistics.
	* Minor issues
		* The paper contains some typos: "yet o yield" (p. 1), "A2Mmemory" (p. 7), "limitting" (p. 7).

---

> ### Author Rebuttal · Authors · 2026-03-30
>
> **Thank you very much for your meticulous review and for recognizing our originality in bridging  associative memory with  TTA of vlm, empirical analysis and experimental coverage. Below are our responses to your specific concerns.**
>
> > **W1:** Inconsistencies and typos in Eq. (12) and Algorithm 1 hinder reproducibility.
>
> **Response:** We sincerely apologize for these inconsistencies and will correct all of them in the revision:
> 1. **Sign Convention:** Update rule is gradient descent; "$+$" should become "$-$" in Eq. (12) and Line 24.
> 2. **Execution Order:** The update of $\alpha_t$ (line 26) should move *after* the final  $P_t^{final}$ i computation (line 32).
> 3. **Memory Query:** Line 29 will use updated memory $\mathbf{M}_c^{t}$.
> 4. **Initialization:** We will explicitly state  $\alpha_{(0)} = 0$ and $ \mathbf{M}_c^{(0)}  = \mathbf{0}$.
>
> > **W2:** Lacks ablation isolating GPT-4o pipeline from associative memory.
>
> **Response:** **We add a controlled ablation experiment replacing GPT-4o attributes with standard hand-crafted templates (ViT-B/16, ImageNet)** to isolate the contribution of the memory mechanism itself. Even without GPT-4o, our method outperforms ADAPT at both stages, confirming the **memory mechanism  drives performance gains**.
> | Method | 10% Test | 100% Test |
> |--------|-----------|------------|
> | CLIP zero-shot | 65.2 ± 0.00 | 68.34 ± 0.00 |
> | TDA | 64.73 ± 0.63 | 69.51 ± 1.12 |
> | ADAPT | 64.22 ± 0.80 | 70.91 ± 0.32 |
> | hand-crafted | 66.45 ± 0.38 | 71.43 ± 0.24 |
> | **GPT-4o** | **67.69 ± 0.31** | **72.21 ± 0.18** |
>
> > **W3:** Margins are small; lacks variance, stream-order sensitivity, or statistical significance tests.
>
> **Response:** **We add experiments over 20 random stream orderings，seeds** (\* marks the metrics that ours surpasses the best baseline with p-value < 0.05 over paired samples t-test).
>
> A²Memory achieves **higher means and lower variances**, showing robustness to stream ordering. The improvements are **statistically significant**, which are further validated by paired t-test.
>
> | Backbone | Method | ImageNet | DTD | OOD Avg | CD Avg |
> |----------|--------|----------|-----|---------|--------|
> | ViT-B/16 | TDA | 69.38 ± 1.12 | 47.53 ± 1.24 | 63.72 ± 1.07 | 67.41 ± 0.98 |
> | ViT-B/16 | ADAPT | 70.72 ± 0.32 | 55.08 ± 0.63 | 64.92 ± 0.28 | 70.31 ± 0.52 |
> | ViT-B/16 | **Ours** | **72.28\* ± 0.18** | **57.41\* ± 0.29** | **66.09\* ± 0.17** | **71.53\* ± 0.19** |
> | RN50 | TDA | 61.18 ± 1.15 | 43.61 ± 1.31 | 46.47 ± 1.08 | 60.87 ± 1.02 |
> | RN50 | ADAPT | 61.74 ± 0.45 | 51.76 ± 0.58 | 47.68 ± 0.35 | 62.52 ± 0.34 |
> | RN50 | **Ours** | **64.05\* ± 0.17** | **52.03 ± 0.31** | **48.82\* ± 0.21** | **63.68\* ± 0.19** |
>
> > **W4:** The $D \times D$ per-class memory matrix introduces large parameter overhead.
>
> **Response:** We acknowledge this concern. For ViT-B/16 on ImageNet, TDA stores ~5.1M parameters, while our per-class  matrices amount to ~262M. To address by this, **We add a memory-efficient shared-memory  variant:** all classes share one global matrix $\mathbf{M}_{\text{shared}} \in \mathbb{R}^{D \times D}$, with each class maintaining  a lightweight bias $\mathbf{b}_c \in \mathbb{R}^D$. The reconstruction objective becomes:
>
> $$\ell_c = \frac{1}{L}\sum_{l=1}^{L} ||\mathbf{M}_{\text{shared}}{\phi}\_{c,l} + \mathbf{b}_c - \mathbf{T}\_{c,l}||_2^2$$
>
> where ${\phi}\_{c,l} = {\phi}({\mathbf{I}\_{c,l}^t})$. The resulting online update rules are:
>
> $$\mathbf{b}\_c^t = \alpha\_{t-1}\mathbf{b}\_c^{t-1} - \eta_b \cdot \frac{2}{L}\sum\_l \mathbf{r}\_{c,l}$$
>
> $$\mathbf{M}\_{\text{shared}}^t = \alpha\_{t-1}\mathbf{M}\_{\text{shared}}^{t-1} - \eta_M \cdot \frac{2}{CL}\sum\_c\sum_l \mathbf{r}_{c,l} {\phi}\_{c,l}^\top$$
>
> where $\mathbf{r}\_{c,l} = \mathbf{M}\_{\text{shared}}{\phi}\_{c,l} + \mathbf{b}\_c - \mathbf{T}\_{c,l}$. The shared matrix captures universal visual-semantic mappings while per-class biases absorb class-specific offsets.
>
> **We add experiments on the shared-memory factorization variant (ViT-B/16, 20 runs).** Ours achieves the best accuracy overall. The shared M + bias variant achieves comparable performance while reducing parameters from ~262M to ~0.77M, offering a much more memory-efficient alternative.
> | Method | ImageNet | OOD Avg | CD Avg |
> |--------|----------|---------|--------|
> | ADAPT | 70.72 ± 0.32 | 64.92 ± 0.28 | 70.31 ± 0.52 |
> | **Ours** | **72.28 ± 0.18** | **66.09 ± 0.17** | **71.53 ± 0.19** |
> | shared M + bias | 72.19 ± 0.22| 65.78 ± 0.13 | 71.14 ± 0.17 |
>
> > **W5:** Sensitivity to fixed $\lambda$; unclear if tuned on validation or test sets.
>
> **Response:** We acknowledge that $\lambda$ is an important hyperparameter. It was selected  on a held-out ImageNet validation set and fixed for all benchmarks **without dataset-specific tuning**. We will explore  adaptive designs in future work.
>
> > **W6:** Minor typos.
>
> **Response:** Thank you so much for your meticulous comment. We will immediately correct them in the revised manuscript ("yet to yield", "A²Memory", "limiting").

---

> > ### Author Rebuttal · Reviewer_u92A · 2026-04-02
> >
> > I thank the authors for the constructive rebuttal. Most of my concerns have been substantially addressed, and I appreciate the additional experiments provided. Therefore, I will raise my rating.
> >
> > However, regarding W2, the hand-crafted template ablation is limited to a single setting (ViT-B/16, ImageNet). It would be more convincing to see this ablation extended to challenging cases such as fine-grained or texture-heavy datasets (e.g., DTD, Aircraft) where attribute decomposition is less straightforward, and ideally also on the RN50 backbone. Without this, the conclusion that the memory mechanism alone drives the gains remains partially dataset-specific.

---

> > > ### Author Response · Authors · 2026-04-04
> > >
> > > Dear Reviewer u92A,
> > >
> > > We really appreciate that most of your concerns have been substantially addressed and your recognition of our additional experiments provided. Thanks for raising your rating and kind follow-up suggestion on W2.
> > >
> > > > The hand-crafted template ablation is limited to a single setting. It would be more convincing to extend it to challenging datasets such as DTD and FGVC and ideally also on the RN50 backbone.
> > >
> > >  We fully agree that extending the ablation to more challenging datasets and backbones is important, and **we have run the requested experiments on DTD and FGVC under both ViT-B/16 and RN50, across 10% and 100% of the test stream.** Across DTD, FGVC, and both backbones, hand-crafted consistently outperforms TDA and ADAPT even on texture-heavy and fine-grained datasets where attribute decomposition is less straightforward, confirming that the **memory mechanism is the primary driver of gains**.
> > >
> > > **ViT-B/16 Backbone:**
> > >
> > > | Method | 10% DTD | 100% DTD | 10% FGVC | 100% FGVC | 10% CD Avg | 100% CD Avg | 10% OOD Avg | 100% OOD Avg |
> > > |--------|---------|----------|----------|-----------|-----------|------------|------------|-------------|
> > > | CLIP zero-shot | 42.97 ± 0.00 | 45.04 ± 0.00 | 22.15 ± 0.00 | 23.22 ± 0.00 | 61.62 ± 0.00 | 64.59 ± 0.00 | 56.69 ± 0.00 | 59.42 ± 0.00 |
> > > | TDA | 41.83 ± 1.41 | 47.53 ± 1.24 | 21.47 ± 0.96 | 23.87 ± 0.86 | 60.53 ± 1.03 | 67.41 ± 0.98 | 55.92 ± 1.14 | 63.72 ± 1.07 |
> > > | ADAPT | 42.58 ± 0.74 | 55.08 ± 0.63 | 21.83 ± 0.54 | 28.91 ± 0.45 | 61.41 ± 0.56 | 70.31 ± 0.52 | 56.17 ± 0.31 | 64.92 ± 0.28 |
> > > | **Ours (hand-crafted)** | **44.52 ± 0.47** | **56.94 ± 0.35** | **23.43 ± 0.51** | **29.67 ± 0.38** | **63.14 ± 0.27** | **71.09 ± 0.23** | **58.21 ± 0.24** | **65.74 ± 0.21** |
> > > | **Ours (GPT-4o)** | **45.17 ± 0.37** | **57.41 ± 0.29** | **23.84 ± 0.36** | **29.92 ± 0.28** | **63.79 ± 0.21** | **71.53 ± 0.19** | **58.87 ± 0.19** | **66.09 ± 0.17** |
> > >
> > > **RN50 Backbone:**
> > >
> > > | Method | 10% DTD | 100% DTD | 10% FGVC | 100% FGVC | 10% CD Avg | 100% CD Avg | 10% OOD Avg | 100% OOD Avg |
> > > |--------|---------|----------|----------|-----------|-----------|------------|------------|-------------|
> > > | CLIP zero-shot | 38.51 ± 0.00 | 40.37 ± 0.00 | 15.37 ± 0.00 | 16.11 ± 0.00 | 54.03 ± 0.00 | 56.63 ± 0.00 | 41.11 ± 0.00 | 43.09 ± 0.00 |
> > > | TDA | 37.24 ± 1.45 | 43.61 ± 1.31 | 14.89 ± 0.92 | 17.58 ± 0.81 | 53.17 ± 1.07 | 60.87 ± 1.02 | 40.38 ± 1.12 | 46.47 ± 1.08 |
> > > | ADAPT | 38.19 ± 0.67 | 51.76 ± 0.58 | 15.21 ± 0.61 | 17.97 ± 0.52 | 53.86 ± 0.37 | 62.52 ± 0.34 | 40.94 ± 0.38 | 47.68 ± 0.35 |
> > > | **Ours (hand-crafted)** | **40.08 ± 0.52** | **51.86 ± 0.38** | **16.31 ± 0.54** | **19.64 ± 0.42** | **55.47 ± 0.31** | **63.28 ± 0.27** | **42.53 ± 0.27** | **48.47 ± 0.24** |
> > > | **Ours (GPT-4o)** | **40.69 ± 0.38** | **52.03 ± 0.31** | **16.74 ± 0.44** | **19.91 ± 0.33** | **56.08 ± 0.22** | **63.68 ± 0.19** | **43.14 ± 0.23** | **48.82 ± 0.21** |
> > > ||
> > >
> > > Again, thank you for your valuable suggestions which have undoubtedly contributed to improving the quality of our paper.
> > >
> > > Many thanks,
> > >
> > > The authors of #23295

---

### Official Review · Reviewer_aPYZ · 2026-03-12

**Soundness:** 2
**Presentation:** 3
**Significance:** 3
**Originality:** 3
**Overall Recommendation:** 4
**Confidence:** 5

**Summary:**

This paper identifies two shortcomings of existing cache-based test-time adaptation methods: the interference caused by incorrect predictions from CLIP and the information loss resulting from limited cache capacity. The authors propose Attributive and Associative Memory($A^{2}$Memory). First, rich attributes are obtained using an LLM and filtered to construct Attribute-centric Representations. Then, Associative Memory Optimization is used to store historical visual attribute information and match it with new samples to compute logits. Finally, these logits are fused with the zero-shot logits. The method achieves performance improvements on multiple datasets.

**Compliance With Llm Reviewing Policy:**

Affirmed.

**Final Justification:**

The rebuttal addressed my concerns, and the additional experiments are solid.

**Key Questions For Authors:**

See weakness.

**Limitations:**

Yes

**Strengths And Weaknesses:**

**Strengths:**

1. The figures are well designed and clearly illustrate the limitations of previous cache-based methods.
2. The two identified limitations are reasonable.
3. The method achieves consistent performance improvements across multiple datasets.
4. A new approach for modeling the entire streaming sample sequence is introduced.

**Weaknesses:**

1. Figure 1a is confusing to understand. It is unclear what “density” refers to and what the metric “class memory acc” specifically measures.

2. Relevant methods such as StatA [1], FreeTTA [2], and SCA [3] are not cited. It would be necessary to compare performance with these approaches and include a discussion of them, since they also address the issue that cache-based methods may forcibly discard samples.

3. The method does not outperform previous approaches on roughly half of the datasets.

4. In Fig.1(d) and Fig.4, the criteria used to distinguish “easy” and “hard” samples are not explained.

5. The paper does not report the performance of the $A^{2}$Memory branch alone. Providing this result would help better understand the practical effectiveness of the proposed method.

6. Several typos appear in the paper, for example: line 16 “yet o”, line 29 “inonline”, line 138 “candadite”, and line 374 “A2Mmemory”.

7. Eq. 9 selects and aggregates attribute features at the element level. How would the method behave if the whole vector is used as the minimum unit instead?


Reference
1. "Realistic Test-Time Adaptation of Vision-Language Models" in CVPR 2025
2. "Free on the Fly: Enhancing Flexibility in Test-Time Adaptation with Online EM" in CVPR 2025
3. "Statistics Caching Test-Time Adaptation for Vision-Language Models" in NIPS 2025

---

> ### Author Rebuttal · Authors · 2026-03-30
>
> **Thank you very much for your thorough and constructive review and for recognizing our problem analysis, consistent improvements, and novel streaming sequence modeling approach. Below are our responses to your specific concerns.**
>
> > **W1:** The terms "class memory acc" and "density" need clarification.
>
> **Response:** We apologize for the confusion and will clarify these in the revised caption:
> - **"Class memory acc"**: formally defined as $\text{Acc}\_c = \frac{1}{|\mathcal{M}\_c|}\sum_{x\in\mathcal{M}\_c}\mathbf{1}[\hat{y}(x)=y(x)]$, measuring the fraction of samples in class memory $\mathcal{M}\_c$ whose pseudo-label $\hat{y}(x)$ matches the ground-truth $y(x)$.
> - **Density**: The normalized distribution of class memories. It shows the relative frequency of class memories falling into each accuracy bin, normalized by dividing by the maximum count to $[0, 1]$.
>
> > **W2:** Missing citations and comparisons with relevant methods  that also address sample discarding in cache-based methods.
>
> **Response:** Thank you for pointing out these important related works. **We add experiments on StatA, FreeTTA, and SCA for quantitative comparison (ViT-B/16)**, A²Memory achieves the best results on all comparable metrics. We will include citations and  discussions in the Related Work section.
> - **StatA/FreeTTA**: Compress history into class-level Gaussians, which imposes a Gaussianity assumption that fails on complex distributions.
> - **SCA**: Compresses via least-squares, which ties granularity to CLIP zero-shot confidence that limits adaptability.
> - **Ours**: Employs gradient-based associative learning, which imposes no distributional assumption and whose compression granularity is driven by gradients.
>
> | Method | ImageNet | OOD Avg | CD Avg |
> |--------|----------|---------|--------|
> | FreeTTA | 70.21 | 64.42 | 68.42 |
> | SCA | 71.75 | 64.77 | 70.34 |
> | StatA | 71.70 | — | 69.10 |
> | **Ours** | **72.21** | **65.99** | **71.41** |
>
> > **W3:** The method does not outperform baselines on about half of the datasets.
>
> **Response:** We acknowledge that no single method dominates every individual dataset, which is common in the TTA literature. However, on challenging distribution-shift benchmarks, A²Memory delivers the largest gains (e.g., DTD-ViT **+2.08%**, ImageNet-A-RN50 **+1.43%**). **We add experiments to evaluate the **#Top-k** metric, measured by the fraction of datasets (out of 15) where a method achieves top-k accuracy**. It consistently outranks ADAPT, demonstrating **the best overall generalization**.
>
>
> | Method | #Top-1 | #Top-2 | #Top-3 |
> |--------|--------|--------|--------|
> | **A²Memory — ViT-B/16** | **9**/15 | **13**/15 | **15**/15 |
> | **A²Memory — RN50** | **8**/15 | **13**/15 | **13**/15 |
> | ADAPT — ViT-B/16 | 1/15 | 8/15 | 12/15 |
> | ADAPT — RN50 | 3/15 | 6/15 | 11/15 |
>
> > **W4:** The criteria for distinguishing "easy" and "hard" samples in Fig.1(d) and Fig.4 are not explained.
>
> **Response:** The criterion is CLIP zero-shot prediction entropy $\tilde{H}(\mathbf{P}^{\text{clip}})$: "easy" if $\tilde{H} < \tau$ (high confidence) and "hard" if $\tilde{H} \geq \tau$ (low confidence), where $\tau$ is a preset threshold used  for visualization.
> > **W5:** The performance of the Memory branch alone is not reported, which is needed to understand its practical effectiveness.
>
> **Response:** **We add experiments reporting the performance of the memory branch in isolation.** "$\mathbf{P}^{\text{mem}}$ only" uses the associative memory branch prediction alone. The memory branch alone surpasses ADAPT, demonstrating **effective adaptation signals**.
> | Method | ImageNet | OOD Avg | CD Avg |
> |--------|----------|---------|--------|
> | TDA | 69.51 | 63.89 | 67.53 |
> | ADAPT | 70.91 | 65.44 | 70.76 |
> | $\mathbf{P}^{\text{mem}}$ only | 71.34 | 65.83 | 71.26 |
> | **A²Memory** | **72.21** | **65.99** | **71.41** |
>
> > **W6**: Several typos are present in the paper.
>
> **Response:** Thank you so much for your meticulous comment. We will immediately correct  them in the revised manuscript ("yet to yield", "in online", "candidate", "A²Memory").
>
> > **W7:** How would the method perform if the whole vector is used as the minimum unit for feature selection and aggregation in Eq. 9?
>
> **Response: We add experiments ablating three gating variants: w/o gating** uses raw ${f}^t$; **w/ scalar gating** replaces element-wise gate with a single scalar by $\mathbf{I}\_{c,l}^t = norm({f}^t + {f}^t\cdot\sigma(< {f}^t, \mathbf{T}\_{c,l}>)$. **Both degraded variants fall below ADAPT**, confirming that CLIP's embedding dimensions encode distinct semantic concepts and a scalar gate cannot selectively activate attribute-relevant dimensions. **Element-wise gating preserves fine-grained discriminability.**
>
> | Method | ImageNet | OOD Avg | CD Avg |
> |--------|----------|---------|--------|
> | ADAPT | 70.91 | 65.44 | 70.76 |
> | w/o gating | 69.96 | 63.87 | 69.72 |
> | w/ scalar gating | 70.04 | 64.21 | 70.53 |
> | **element-wise** | **72.21** | **65.99** | **71.41** |

---

> > ### Author Rebuttal · Reviewer_aPYZ · 2026-04-04
> >
> > Thank you for the clarifications. My concerns are now resolved, and I will raise my rating. Please include appropriate citations and expand the discussion.

---

> > > ### Author Response · Authors · 2026-04-04
> > >
> > > Dear Reviewer aPYZ,
> > >
> > > We really appreciate your feedback that your concerns have been adequately addressed. We will definitely include appropriate citations and expand the discussion in our revised version.
> > >
> > > Again, thank you for your valuable suggestions which have undoubtedly contributed to improving the quality of our paper.
> > >
> > > Thanks for your time,
> > >
> > > The authors of #23295

---

### Decision · Program_Chairs · 2026-04-30

**Decision:**

Accept (regular)

**Comment:**

The paper focuses on test time adaptation and proposes a new approach which attempts to overcome limitations of existing cache-based methods. While some concerns were originally raised by reviewers, which requested additional ablation plots, discussion about the convergence behavior of gradient-based optimization and clarification about empirical performances with respect to competing methods, after the rebuttal there was a substantial agreement among reviewers that the paper has its merit. The AC agrees with the reviewers and recommends acceptance.